# Astrocyte-Secreted Lcn2 Modulates Dendritic Spine Morphology

**DOI:** 10.3390/cells14030159

**Published:** 2025-01-21

**Authors:** Marta Doliwa, Bozena Kuzniewska, Karolina Nader, Patryk Reniewicz, Leszek Kaczmarek, Piotr Michaluk, Katarzyna Kalita

**Affiliations:** Laboratory of Neurobiology, Nencki-EMBL Partnership for Neural Plasticity and Brain Disorders-BRAINCITY, Nencki Institute of Experimental Biology, Polish Academy of Sciences, 3 Pasteur Street, 02-093 Warsaw, Poland

**Keywords:** Lcn2, cLTP, dendritic spines, astrocytes, exocytosis, glia, gliotransmitter, aberrant plasticity

## Abstract

Learning and memory formation rely on synaptic plasticity, the process that changes synaptic strength in response to neuronal activity. In the tripartite synapse concept, molecular signals that affect synapse strength and morphology originate not only from the pre- and post-synaptic neuronal terminals but also from astrocytic processes ensheathing many synapses. Despite significant progress made in understanding astrocytic contribution to synaptic plasticity, only a few astrocytic plasticity-related proteins have been identified so far. In this study, we present evidence indicating the role of astrocyte-secreted Lipocalin-2 (Lcn2) in neuronal plasticity. We show that Lcn2 expression is induced in hippocampal astrocytes in a kainate-evoked aberrant plasticity model. Next, we demonstrate that chemically induced long-term potentiation (cLTP) similarly increases Lcn2 expression in astrocytes of neuronal–glial co-cultures, and that glutamate causes the immediate release of Lcn2 from these cultures. Additionally, through experiments in primary astrocytic cultures, we reveal that Lcn2 release is triggered by calcium signaling, and we demonstrate that a brief treatment of neuronal–glial co-cultures with Lcn2 alters the morphology of dendritic spines. Based on these findings, we propose Lcn2 as an activity-dependent molecule released by astrocytes that influences dendritic spine morphology.

## 1. Introduction

Learning new information and storing it as a memory are among the brain’s major functions. At the cellular level, these cognitive processes rely mainly on synaptic plasticity, which is the ability of synapses to either strengthen or weaken in response to activity [1]. To date, synaptic plasticity has been studied mainly on the excitatory synapses that are located primarily on dendritic spines—small dendritic protrusions [2]. The spines exhibit a continuum of shapes that are commonly categorized into mushroom, long, and stubby [3,4]. The morphological characteristics of dendritic spines, such as the size of the spine head and neck, have been correlated with synapse strength [5,6]. One of the most studied models of synaptic plasticity is long-term potentiation (LTP), where, upon stimulation, synapses are strengthened by the insertion of AMPA (α-amino-3-hydroxy-5-methyl-4-isoxazolepropionic acid) receptors into a spine membrane; this process is also associated with the spine enlargement [7,8,9,10]. Since its discovery, the mechanisms underlying LTP induction, expression, and maintenance have been described in detail [7,8,9,10].

Recently, however, a more complex model of synapse and its plasticity has gained significant attention. The tripartite synapse model appreciates not only signals originating in the pre- or post-synaptic terminals of neurons but also those arising from fine astrocytic processes that ensheathe a substantial proportion of synapses in the central nervous system [11,12,13,14]. The astrocytes sense various neurotransmitters, including glutamate, ATP, GABA (γ-aminobutyric acid), norepinephrine, and many others [15,16,17], and respond by increasing intracellular Ca^2+^ levels. Ca^2+^ signals have been shown to trigger the exocytosis of gliotransmitters, such as D-serine, ATP, or glutamate, which in turn influences synaptic strength and plasticity [13,18,19,20]. Although significant progress has been made in the understanding of astrocytic modulation of synaptic transmission, ongoing efforts aim to identify the novel activity-dependent molecules released by astrocytes and clarify the mechanisms of their action on neuronal cells.

One of the proteins that recently emerged as an important mediator of neuronal morphology and function, which has a glial origin, is Lipocalin-2 (Lcn2, also known as NGAL and 24p3) [21,22,23,24,25]. It is a small, extracellular protein that binds siderophore-complexed iron and contributes to the innate immune response by limiting iron availability for pathogens [26,27]. In the central nervous system, Lcn2 has primarily been studied in the context of brain pathologies, including neuroinflammation, ischemia, pain, multiple sclerosis, epileptogenesis, stress, and Alzheimer’s disease [21,23,24,25,28,29]. Data collected from different models consistently show increased Lcn2 expression in the brain in response to disease or injury. Depending on the model and the brain region, Lcn2 has been detected in various cell types, including astrocytes, microglia, neurons, epithelial cells, and choroid plexus endothelial cells [21,23,25,28,30,31,32,33]. However, the precise physiological functions of Lcn2 remain elusive. Research using mice with constitutive knockout of Lcn2 revealed that the absence of Lcn2 impairs hippocampal neurogenesis [34] and alters neuronal morphology in the hippocampus and amygdala, with varying effects across neuron subpopulations and brain structure [22,24,25]. These morphological changes have been reported to be accompanied by functional impairments, such as decreased LTP in the CA1 region of the dorsal hippocampus and increased excitability of pyramidal neurons of the hippocampal CA1 and basolateral amygdala [22,24,25]. Moreover, Lcn2 knockout mice exhibit anxiety, depression-like behaviors, and mild cognitive impairment [22]. These observations underscore the need for a more comprehensive understanding of the role of Lcn2 in brain function.

In the present study, we demonstrate in vivo that the expression of Lcn2 in astrocytes increases during kainate-induced aberrant plasticity. Furthermore, we employ cell cultures, pharmacological manipulations, molecular biology techniques, and live-cell imaging to demonstrate in vitro the following: (1) Lcn2 is expressed in astrocytes in response to chemical LTP (cLTP); (2) the release of Lcn2 can be triggered by glutamate stimulation; (3) the release of Lcn2 is induced by calcium influx to astrocytes; (4) the Lcn2 treatment induces rapid changes in dendritic spine morphology. Together, our findings suggest the involvement of astrocytic Lcn2 in the regulation of dendritic spine morphology.

## 2. Materials and Methods

### 2.1. Animals

Male 6-month-old C57BL/6 mice were maintained in the Animal House of the Nencki Institute of Experimental Biology and were treated in compliance with the ethical standards of European and Polish regulations. Animals were housed individually with free access to food and water, at 23–25 °C, under a 12 h light/12 h dark cycle. The experimental procedures were approved by the 1st Local Ethics Committee in Warsaw (permission no. 678/2015).

### 2.2. Kainic Acid Injections

Mice were anesthetized by isoflurane inhalation (Baxter, Unterschleißheim, Germany) in oxygen. Then, they were put in a stereotaxic frame and injected unilaterally with kainic acid (70 nL of 20 mM solution; Nanocs, New York, NY, USA) into the left CA1 area of the dorsal hippocampus (coordinates relative to bregma: AP −1.8 mm, ML +1.7 mm, and DV −2.3 mm). During the procedure, the mice were kept on a heating pad to prevent hypothermia. Control animals used for the immunohistochemistry staining were injected with NaCl following the procedure described above.

### 2.3. Staining of Brain Slices

Eighteen days after kainate injection, mice underwent transcardial perfusion with 4% paraformaldehyde in PBS. The brains were post-fixed in the same solution overnight (ON) at 4 °C and sectioned on a vibratome into 40 μm-thick slices.

The brain sections used for Nissl staining were air-dried on slides and stained with 0.1% cresyl violet solution (in 3% acetic acid) for 5 min. Following washing, the sections were dehydrated, cleared in xylene, and closed with coverslips.

The brain slices used for immunohistochemistry staining were washed thrice with PBS, and the endogenous peroxidase was quenched for 5 min (10% methanol and 3% H_2_O_2_ in PBS). After being washed thrice with PBS, slices were permeabilized in 0.2% Triton X-100 (BioShop, Burlington, ON, Canada) in PBS for 15 min, washed thrice with PBS, blocked in 10% normal horse serum (NHS) in PBS at RT, and incubated ON at 4 °C with primary anti-Lcn2 antibody (AF1857 (R&D, Minneapolis, MN, USA), diluted 1:200 in PBS with 2% NHS). Then, they were washed three times with PBS and incubated for 2 h at RT with a secondary biotinylated anti-goat antibody (Vector Laboratories (Newark, CA, USA), BA-9500, diluted 1:500 in PBS with 3% NHS). After being washed thrice with PBS, the cells were incubated for 1.5 h with Avidin/Biotin Complex (ABC) Reagent (Vector Laboratories, (Newark, CA, USA), PK-6100, VECTASTAIN^®^ Elite^®^ ABC-HRP Kit, Peroxidase), washed twice with PBS, incubated for 1–2 min with SIGMAFAST™ DAB (Sigma-Aldrich, Saint Louis, MO, USA) and washed with milli-Q water. Then, the slices were dried on slides, dehydrated with ethanol, cleared in xylene, mounted with DEPEX, and imaged under an Olympus IX70 microscope (Olympus Corporation, Tokyo, Japan).

For immunocytochemistry staining brain slices were washed with a TNT buffer (100 mM Tris pH 7.4, 150 mM NaCl, 0.05% Tween 20), permeabilized in 0.3% Triton X-100 (BioShop, Burlington, ON, Canada) in TNT for 10 min, washed with TNT, and blocked for 1 h in a TNB buffer [0.1 M Tris-HCl pH 7.5, 0.15 M NaCl, 0.5% blocking reagent (PerkinElmer, FP1020, Shelton, CT, USA)]. After blocking, slices were incubated ON at 4 °C with primary antibodies diluted in a TNB buffer: anti-mouse-Lcn2 antibody (AF-1857, 1:300, goat, R&D, Minneapolis, MN, USA) and anti-GFAP-Cy3 (C9205, 1:1000, mouse, Sigma-Aldrich, Saint Louis, MO, USA). Next, the slices were washed with TNT, incubated for 1 h at RT with secondary biotinylated anti-goat antibody (Vector Laboratories, Newark, CA, USA, BA-9500, diluted 1:500 in a TNB buffer), washed with TNT, and incubated for 1 h at RT with Alexa-Avidin (1:500 in TNB). After washing with TNT, slices were mounted on slides and imaged under a Leica TCS SP5 confocal microscope (Leica, Wetzlar, Germany) equipped with HCX PL APO CS 40.0 × 1.25 oil immersion objective. Acquired images had 1024 × 1024 resolution and 0.378 µm × 0.378 µm pixel size.

### 2.4. DNA Vectors

To create the pβactin_Lcn2_SEP plasmid: (1) the *Lcn2* gene was amplified from P14 rat hippocampal cDNAs sample using Fwd: 5′-AGGAAGCTTGCTGAAACCATGGGCCTGGG, Rev: 5′-GGGAATTCGTTGTCAATGCATTGGTCGGTGGG primers. The sequence of amplified *Lcn2* gene was 100% identical to the sequence deposited in GeneBank (NM_130741.2). The *Lcn2* gene was cloned to pβactin_GFP at HindIII, EcoRI restriction sites yielding the pβactin_Lcn2_GFP plasmid (2) the *superecliptic pHluorin* (*SEP*) was amplified using a pair of primers: Fwd: 5′-GGGAATTCAAAGGAGAAGAACTTTTCACTGGAG and Rev: 5′-CCTCTAGATCATTTGTATAGTTCATCCATGCC (3) *green fluorescent protein (GFP)* fragment was cut out from the pβactin_Lcn2_GFP plasmid with EcoRI and XbaI restriction enzymes, and the amplified *SEP* gene was cloned at this restriction sites. To create the pZac2.1_gfa_ABC1D__Lcn2_FLAG_SEP: (1) the *Lcn2_SEP* sequence was amplified from pβactin_Lcn2_SEP using Fwd: 5′-CGCGCTAGCGCTAAGCTTGCTGAAAC, Rev: 5′-GGCGCTCGAGGCGGCCGCATTCATTTGTATAG and cloned to pZac2.1gf_ABC1D__MCS plasmid at NheI, XhoI restriction sites yielding the pZac2.1gf_ABC1D__Lcn2_SEP plasmid (2) the FLAG linker was cloned in-between Lcn2 and SEP coding sequences using NEBuilder HiFi DNA Assembly Kit (New England Biolabs, Ipswich, MA, USA), according to the manufacturer protocol. The plasmid backbone was amplified from pZac2.1gf_ABC1D__Lcn2_SEP with Q5 polymerase used alongside with Q5 High GC Enhancer and Fwd: 5′-AAAGGAGAAGAACTTTTCACTGG and Rev: 5′-GTTGTCAATGCATTGGTCGGTGGG primers. The single-stranded oligonucleotide that was cloned into the backbone contained FLAG sequence and plasmid overlaps necessary for the assembly (5′-TCCCACCGACCAATGCATTGACAACGGATCCGACTACAAGGATGACGATGACAAGAGTAAAGGAGAAGAACTTTTCACTGGAG). After each cloning step, the cloned gene was sequenced to confirm the absence of mutations that could potentially be introduced by PCR.

### 2.5. Cell Culture and Transfection

Dissociated primary hippocampal and cortical cultures were prepared from P0 Wistar rats as described previously [35,36]. For the life cell imaging experiment, hippocampal cells were transfected at DIV 7–9 with 2 µg of pSynapsin_GFP plasmid. Transfection was performed for 1 h in a medium containing Neurobasal-A without phenol red (Gibco/Life Technologies, Waltham, MA, USA) and 1% GlutaMAX™ (Gibco/Life Technologies, Waltham, MA, USA) and was carried out using Lipofectamine 2000 reagent (Thermo Fisher Scientific, Waltham, MA, USA). After transfection, the conditioned maintenance medium was recovered to the cells. For the glutamate stimulation, the dissociated cortical cells were electroporated with 1.5 µg of pβactin_Lcn2_SEP and 1.5 µg of pβactin_RFP using Rat Neuron Nucleofector Kit (Amaxa™, Lonza, Basel, Switzerland), according to the protocol provided by the manufacturer.

The primary astrocytic cultures were prepared from P0 Wistar rats. The cortices were isolated and dissociated following the same procedure as used for neuronal–glial co-cultures [36]. For the astrocytic cultures, the plating medium (PM) comprised DMEM (high glucose, with GlutaMAX™ Supplement), 10% FBS, 1% penicillin, and streptomycin (all from Gibco/Life Technologies, Waltham, MA, USA). After isolation, the cells suspended in PM were transferred to 75 cm^2^ culture flasks. The fresh PM medium was provided the day after plating. After reaching 80% confluency, the cells were shaken (6 h, 240 rpm, 37 °C) in fresh PM to remove the oligodendrocytes. Afterward, the cells were washed with PM, and the PM medium was exchanged. On the following day, the cells were shaken (1 h, 180 rpm, 37 °C) to remove microglial cells. Then, the astrocytes were washed with PM, and the fresh PM medium was provided. For the experiments, the astrocytic cells were electroporated with 5 µg of pZac_gfa_ABC1D__Lcn2_FLAG_SEP plasmid using nucleofection reagents for primary mammalian glial cells, according to the manufacturer’s protocol (Amaxa™, Lonza, Basel, Switzerland).

### 2.6. Glutamate Stimulation

At DIV 7–8, the culture medium from pβactin_Lcn2_SEP transfected cortical cultures was collected. Then the cultures were stimulated with 50 µM glutamate (Sigma-Aldrich, Saint Louis, MO, USA) either for 10 min or for 30 min at 37 °C in a fresh culture medium (Neurobasal-A without phenol red, 2% B27, 1% GlutaMAX™ [all from Gibco/Life Technologies, Waltham, MA, USA]). The unstimulated cells were kept for 30 min at 37 °C in a fresh culture medium without glutamate addition. The culture medium before and after glutamate treatment was tested for the presence of the Lcn2–SEP protein using the standard western blot procedure. To reduce background, the membrane was blocked overnight (ON) at 4 °C with 5% non-fat milk in Tris-buffered saline with 0.1% Tween^®^-20 Detergent (BioShop, Burlington, ON, Canada) (TBST). The primary anti-Lcn2 antibody (R&D, Minneapolis, MN, USA, AF-3508, ON at 4 °C) was diluted 1:200, and the secondary antibody was diluted 1:10000 (anti-goat HRP, Invitrogen 81-1620; Abingdon, UK), 1 h at room temperature (RT)). The signal was detected with Ameshram ECL Prime chemiluminescent substrate (Cytiva, Marlborough, MA, USA), and the quantification was carried out using Image Lab 6.1 software (BioRad, Hercules, CA, USA). To avoid confounding effects from unequal Lcn2–SEP overexpression between wells, we have a standardized level of Lcn2–SEP released after treatment relative to its basal level in a culture medium before stimulation.

### 2.7. Life Cell Imaging

Life cell imaging was carried out at DIV 19–23 on the primary hippocampal cultures transfected with pSynapsin_GFP. The cells were placed in the imaging chamber, which was heated to 37 °C and saturated with 5% CO_2_, and imaged with a Leica TCS SP8 (Leica, Wetzlar, Germany) confocal microscope using HC PL APO CS2 63×/1.20 water immersion objective and 488 laser line. Acquired images of secondary and tertiary dendrites had 1024 × 1024 resolution and 0.069 × 0.069 µm pixel size. The same dendritic spines were imaged at two distinct time points (0 min and 20 min). At “time 0”, 500 ng of rat recombinant Lcn2 (R&D, Minneapolis, MN, USA, 3508-LC-050) dissolved in 50 μL of the conditioned culture medium was added to the “Lcn2” sample. In the “control” sample, the culture medium was added instead of the recombinant Lcn2. The morphological parameters of the dendritic spines (area, length, head_width, length/head_width ratio) were determined using the SpineMagic ver. 023 software (https://github.com/NenckiSpines/SpineMagick, accessed on 10 January 2025) [37]. The spines were then divided into “mushroom”, “stubby”, and “long” groups based on their shape at time 0. The classification was performed using the SpineTools Python script [38]. The classification process involved an automated method where the script first segregates spines into 36 initial clusters. These clusters were then manually refined into three distinct spine types: mushroom spines (characterized by short necks and prominent heads), long spines (with heads smaller relative to the neck), and stubby spines (small spines without heads). We analyzed the spine length/width ratios in two groups: (1) mushroom and stubby spines combined, and (2) long spines. For each group, we normalized their length/width ratio for both times (0 and 20 min) to an average length/width ratio of spines at time 0 (separately for rLcn2-treated group and control). Since the most significant source of variance in our data are individual cells, we have averaged length/width ratio parameters per cell and treated them as an independent variable in our analysis. The data were analyzed using GraphPad Prism (GraphPad Software 10, Boston, MA, USA) with repeated-measures two-way ANOVA and a post hoc Šídák’s multiple comparisons test.

### 2.8. Stimulation of Primary Rat Astrocytic Cultures with Ionomycin and ATP

The experiment was carried out two days after the transfection of primary astrocytic cultures with pZac_gfa_ABC1D__Lcn2_FLAG_SEP. First, the culture medium was collected to check baseline Lcn2 release. Samples of the cell medium were centrifuged for 10 min at 16,000× *g* to remove all cell debris, and the supernatants were mixed with a 5× Leammli buffer and analyzed using the standard western blot procedure. The cells were washed twice with an NRS buffer (10 mM HEPES-NaOH, pH 7.4; 140 mM NaCl, 5 mM KCl, 0.8 mM MgCl_2_, 2 mM CaCl_2_, 10 mM glucose) and then incubated for 5 or 20 min with an NRS buffer containing 50 µM ATP (Merck, Rahway, NJ, USA) or 5 µM ionomycin (Alomone Labs, Jerusalem, Israel). Control samples were incubated with an NRS buffer with 0.025% DMSO. The proteins released to the NRS buffer during stimulation were precipitated at 4 °C with 60% EtOH, centrifuged at 15,000× *g* for 5 min, resuspended in a 1× Leammli buffer, and probed for the presence of Lcn2–SEP using the standard western blot procedure. To reduce background and increase sensitivity of Lcn2 detection, we blocked the membrane with 5% non-fat milk in TBST ON at 4 °C, used low concentrations of primary (anti-Lcn2, R&D, Minneapolis, MN, USA, AF-3508, 1:1000, ON at 4 °C) and secondary antibodies (anti-goat HRP, Invitrogen (Abingdon, UK), 81-1620, 1:100,000, 1 h at RT) and detected signal with SuperSignal™ West Femto chemiluminescent substrate (Thermo Fisher Scientific, Waltham, MA, USA). Signal quantification was carried out using GelAnalyzer 19.1 software (available at www.gelanalyzer.com) by Istvan Lazar Jr. and Istvan Lazar Sr. To avoid confounding effects from unequal Lcn2–SEP overexpression between wells, we have standardized the level of Lcn2–SEP released after treatment relative to its basal level in a culture medium before stimulation.

### 2.9. Chemical Long-Term Potentiation (cLTP)

The cLTP experiment was carried out on dissociated hippocampal cultures at DIV 21–29 based on the previously published protocol [39]. In order to silence the spontaneous synaptic transmission, the cells were incubated for 3 h in the conditioned medium mixed with 40 μM 6-cyano-7-nitroquinoxaline-2,3-dione (CNQX, Sigma-Aldrich, Saint Louis, MO, USA), 100 μM (2*R*)-amino-5-phosphonovaleric acid (APV, Sigma-Aldrich, Saint Louis, MO, USA), and 5 μM nimodipine (Sigma-Aldrich, Saint Louis, MO, USA). Then, the silencing mixture was removed, and the conditioned medium with 50 μM forskolin (Sigma-Aldrich, Saint Louis, MO, USA), 0.1 μM rolipram (Sigma-Aldrich, Saint Louis, MO, USA), and 50 μM picrotoxin (PTX, Sigma-Aldrich, Saint Louis, MO, USA) was added to the cells for 24 h to induce the cLTP. The control cells were silenced for 3 h and then kept for 24 h in a conditioned medium without the drugs.

### 2.10. RNA Isolation and Quantitative Real-Time PCR

For the experiment, the cells subjected to the cLTP procedure were lysed using an RLT buffer (Qiagen, Hilden, Germany), and the RNA isolation was performed with a Qiagen RNeasy mini kit (Qiagen, Hilden, Germany). The DNA was removed from samples using a Turbo DNA-free kit (Invitrogen, Abingdon, UK). The reverse transcription reaction was carried out using SuperScript™ IV Reverse Transcriptase (Invitrogen, Abingdon, UK). The PCR was performed in the Applied Biosystems 7900HT Fast Real-Time PCR System (Waltham, MA, USA) using rat *Lcn2* (Thermo Fisher Scientific, Waltham, MA, USA, Rn00590612_m1), and rat *Gapdh* (Thermo Fisher Scientific, Waltham, MA, USA, Rn01775763_g1) probes and TaqMan Fast Advanced Master Mix (Thermo Fisher Scientific, Waltham, MA, USA). The data were analyzed using ∆∆CT relative quantification method and normalized relative to *Gapdh* levels.

### 2.11. Immunostaining of the Cell Cultures

For immunocytochemistry, the cells subjected to cLTP were fixed with pre-warmed 4% paraformaldehyde (PFA) and 4% sucrose in phosphate-buffered saline (PBS), pH~7.4, washed three times with PBS and kept in PBS with 0.02% sodium azide until the immunostaining. On the day of the experiment, fixed cells were washed three times with PBS, permeabilized with 0.3% Triton X-100 (BioShop, Burlington, ON, Canada) in PBS for 10 min, washed with PBS, and blocked for 1 h at RT using a TNB buffer [0.1 M Tris-HCl pH 7.5, 0.15 M NaCl, 0.5% blocking reagent (PerkinElmer, Shelton, CT, USA, FP1020)] with 10% normal donkey serum (NDS). Next, the cells were incubated overnight (ON) at 4 °C with anti-Lcn2 antibody (goat, 1:300, R&D, AF-3508), washed 3 times with PBS, and incubated with anti-goat Alexa Fluor 488 (Invitrogen, Abingdon, UK, A-11055) for 1 h at RT. After thrice wash with PBS, they were incubated ON at 4 °C with GFAP (rabbit, 1:500, Abcam, Cambridge, UK, AB7260) and MAP-2 (mouse, 1:500, m1406, Sigma-Aldrich, Saint Louis, MO, USA) antibodies. Next, they were washed three times with PBS and incubated for 1 h at RT with anti-rabbit Alexa Fluor 568 (Invitrogen, Abingdon, UK, A-11036) and anti-mouse Alexa Fluor 647 (Invitrogen, Abingdon, UK, A-31571) antibodies. All antibodies were diluted in a TNB buffer with 10% NDS. After being washed trice with PBS, the cells were mounted using a Fluoromount-G™ Mounting Medium, with DAPI (Thermo Fisher Scientific, Waltham, MA, USA), and kept at 4 °C until imaging. The images were acquired with a Leica TCS SP8 (Leica, Wetzlar, Germany) confocal microscope using HC PL APO CS2 20×/0.75 immersion objective, 488/561/640 nm lines of white light laser, and 405 nm line of pulse diode laser. Four fields were collected from each culture well. Acquired images had 1024 × 1024 resolution and 0.568 × 0.568 µm pixel size. For the analysis, the images were processed using the Fiji distribution of ImageJ software available at https://imagej.net/software/fiji/, accessed on 10 January 2025 [40]. The maximum intensity projections were obtained from the Z-stack series. In the DAPI channel, the displayed pixel values were set to minimum = 0 and maximum = 30. The appropriate channels were merged in order to calculate the number of Lcn2/GFAP, Lcn2/MAP-2, MAP-2/DAPI, and GFAP/DAPI positive cells. Finally, the percentage of Lcn2/GFAP cells relative to GFAP/DAPI was calculated.

### 2.12. Total Internal Reflection Fluorescence (TIRF) Microscopy

For the TIRF imaging, the coverslip with astrocytes was placed in a living chamber heated to 32 °C with a constant flow of NRS buffer. First, the one-minute movie of the unstimulated astrocyte was recorded, and then the NRS buffer was changed to an NRS buffer with 5 µM ionomycin. The one-minute movies of the stimulated cells were recorded two minutes after the start of the buffer exchange. Each movie consisted of 2000 frames recorded at the rate of 33 Hz with 11 millisecond exposure time. The movies were recorded on a QuantEM™ 512SC EMCCD camera (Photometrics, Tucson, AZ, USA) with 2 × 2 binning, resulting in a 256 × 256 frame resolution and 0.32 × 0.32 µm pixel size. The imaging was performed on Carl Zeiss TIRF (Carl Zeiss AG, Oberkochen, Germany) microscope equipped with Alpha Plan-Apochromat 100×/1.46 oil immersion objective and 488 laser line brought to an angle required for total internal reflection microscopy. The presence of the exocytosis events was determined using Suite2p (version 0.7.4.) [41] software and verified using custom ImageJ (available at https://imagej.net/software/fiji/, accessed on 10 January 2025) macros [40].

### 2.13. Statistical Analysis

The statistical analysis was performed using GraphPad Prism software (GraphPad Software ver. 10, Inc., Boston, MA, USA). The two-group comparison was carried out using an unpaired *t*-test with the exception of the quantification of Lcn2–SEP release rate, which was analyzed with a paired *t*-test. The multiple group comparison was made using either one-way analysis of variance (ANOVA) followed by Dunnett’s or Tukey’s multiple comparisons or repeated-measures two-way ANOVA followed by Šídák’s post hoc test. The data were presented as a mean ± standard errors of the means (SEM).

### 2.14. Figures Preparation

We used ImageJ (available at https://imagej.net/software/fiji/, accessed on 10 January 2025) [40], CorelDraw 2014, and Biorender (https://www.biorender.com/) to prepare the figures. GPT-4 (https://chatgpt.com/) and Claude 3.5 Sonnet (https://claude.ai/) were used to improve the readability and grammar of the text, and suggested changes were carefully reviewed and edited to ensure manuscript accuracy and coherence.

## 3. Results

### 3.1. Lipocalin-2 Expression in Astrocytes Is Increased in a Kainate Model of Aberrant Plasticity

Kainate (KA) is a potent analog of glutamate that triggers considerable synaptic plasticity in the dentate gyrus (DG) region of the hippocampus by causing excitotoxicity in CA subfields. It is, therefore, a great model to test the engagement of genes in synaptic plasticity [42]; we used kainate to identify cells expressing Lcn2 during synaptic plasticity. We delivered kainate into three animals through unilateral intrahippocampal injection of kainic acid and looked for changes in Lcn2 expression 18 days after treatment. Nissl staining demonstrated that, as expected, the kainate injections induced cell death in CA1–CA3 subfields and led to structural reorganization of DG as manifested by granule cell dispersion (Figure 1A).

The immunohistochemical analysis of the brain slices revealed that kainate treatment induces Lcn2 expression in DG (Figure 1B). The Lcn2 positive cells resided predominantly in the DG hilus, subgranular zone, as well as in a molecular layer; some Lcn2 positive cells were also present in the granule cell layer (Figure 1B). Subsequent immunocytochemistry demonstrated that kainate increased the expression of Lcn2 in the hilus and granule cell layer of the DG in all three tested animals and that, in this region, the Lcn2 signal localized almost exclusively to GFAP-positive astrocytes (Figure 1C).

### 3.2. Lipocalin-2 Expression in Astrocytes Is Increased After cLTP

Since Lcn2 levels increase during aberrant plasticity in vivo, we decided to determine if Lcn2 expression is also induced in the model of synaptic plasticity—LTP, especially since it has been previously reported that Lcn2-deficient mice have impairments in hippocampal LTP [22]. In order to activate a big population of cells, we applied chemical LTP (cLTP) in primary dissociated neuronal–glial co-cultures [39]. This protocol has been previously shown to lead to morphological changes in dendritic spines, increased network activity, as well as induced gene transcription [35,43,44,45,46,47,48]. To check for *Lcn2* mRNA levels, we used quantitative real-time RT-PCR. The analysis of RNAs isolated from the cultures showed an increase in *Lcn2* mRNA levels 24 h after cLTP induction compared to untreated controls. Interestingly, *Lcn2* mRNA levels 6 h after cLTP were not statistically different from the control (Figure 2A).

In the kainate-induced plasticity experiment (Figure 1), we found the Lcn2 signal to colocalize almost exclusively with astrocytic marker—GFAP. In order to check if the same can be observed in the cLTP model, we performed an immunofluorescent analysis of neuronal–glial co-cultures. The analysis revealed that the Lcn2-positive signal increases 24 h after cLTP induction and colocalizes with some (but not all) GFAP-positive astrocytes (Figure 2B). Interestingly, we were not able to identify the Lcn2 signal in MAP2-positive cells. The weak signal that did not colocalize with GFAP arises from autofluorescence in the 488 channel, which was also present in the control staining without the primary anti-Lcn2 antibody (Appendix A). The quantification of the number of Lcn2 and GFAP double-positive cells has shown an increase in Lcn2 protein levels in response to 24 h cLTP compared to control (Figure 2C). This localized Lcn2 increase, observed only in a subpopulation of astrocytes, was not clearly detectable by western blotting (Appendix A).

### 3.3. Lipocalin-2 Is Released Immediately After Glutamate Stimulation

The astrocytes actively participate in the synaptic plasticity by releasing gliotransmitters in response to activity-dependent neurotransmitter secretion [49,50]. To determine if Lcn2 is released in response to neuronal stimulation, we overexpressed Lcn2 in fusion with pH-dependent superecliptic pHluorin (SEP) in primary neuronal–glial co-cultures and treated them with 50 µM glutamate at 7–8 days in vitro (DIV). As the molecules essential for plastic changes are rapidly released after stimulation, we specifically conducted a western blot analysis of the culture medium at 10 and 30 min after glutamate treatment. The level of the Lcn2 protein increased 10 min after glutamate, while 30 min incubation was not statistically different from not-treated cultures (Figure 3). This result shows that Lcn2 is released upon stimulation into the extracellular space.

### 3.4. Lipocalin-2 Release Is Triggered by the Calcium Influx to the Astrocytic Cells

In the tripartite synapse model, the release of neurotransmitters triggers an increase in cytosolic Ca^2+^ levels in astrocytes, consequently leading to the secretion of gliotransmitters. In order to explore whether the mechanism of Lcn2 exocytosis is Ca^2+^-dependent, we used Lcn2–SEP expression plasmid. Initially, we expressed this construct in primary astrocytic cultures and used western blotting to measure its release 5 and 20 min after treatment with either Ca^2+^ ionophore—ionomycin or ATP. We selected ATP, in addition to directly increasing intracellular Ca^2+^ through ionomycin application, as its action through purinergic receptors is well documented to trigger an increase of Ca^2+^ in astrocytes through the PLC/IP_3_ signal cascade [51,52,53]. Importantly, ATP has previously been shown to increase Ca^2+^ levels in cultured astrocytes and to induce the release of SEP-tagged plasticity protein BDNF in these cultures [54]. To avoid confounding effects from the ATP-induced LTP [55], we carried out experiments in pure astrocytic cultures. The results revealed an increase in released Lcn2 20 min after ATP and ionomycin treatment (Figure 4).

To gain insights into the dynamics of individual vesicle release, we employed total internal reflection microscopy (TIRF). With this technique, we utilized the pH-sensitive nature of SEP to track the release of Lcn2–SEP from primary astrocytic cultures (Figure 5A).

We recorded one-minute movies at baseline and two minutes after initiating the buffer exchange to one with ionomycin. The release of Lcn2–SEP from vesicles was observed as a brief increase in fluorescence, which was followed by diffusion of protein from the plasma membrane, as indicated by the decline in fluorescence (Figure 5C,D). The analysis of the Lcn2 release rate demonstrated an increase in Lcn2 exocytosis 2 min after ionomycin treatment, indicating that Lcn2 secretion from astrocytes is Ca^2+^-inducible (Figure 5B,E).

### 3.5. Lipocalin-2 Induces Rapid Changes in Dendritic Spine Morphology

The structural remodeling of dendritic spines during LTP occurs immediately after neuronal stimulation [56,57]. Therefore, we wanted to check if a transient exposure to Lcn2 can elicit rapid changes in dendritic spine shape. To investigate that, we transfected cultures with GFP, and at 19–23 DIV, we performed confocal live cell imaging to record changes in spine shape occurring 20 min after Lcn2 treatment. Next, we classified spines into mushroom, long, and stubby groups based on their shape at the beginning of the imaging. We observed that a 20 min Lcn2 treatment increased the length-to-width ratio of mushroom and stubby spines (Figure 6B), showing that a short incubation with Lcn2 leads to a change in shape toward a more immature form of spines—narrow and elongated. There were no significant changes observed in the morphology of long spines following 20 min of Lcn2 treatment (Figure 6C), indicating that the effects of Lcn2 do not uniformly affect all spine types.

## 4. Discussion

Our study aims to provide evidence supporting the involvement of astrocyte-secreted Lcn2 in synaptic plasticity. Specifically, we demonstrate the following: (i) the Lcn2 protein is elevated in astrocytes after kainate-induced plasticity; (ii) cLTP increases expression of Lcn2 in neuronal–glial co-cultures; (iii) Lcn2 is released immediately after glutamate-induced synaptic activation; (iv) the astrocytic Lcn2 release is Ca^2+^-dependent; moreover, (v) brief exposure to recombinant Lcn2 induces morphological changes in dendritic spines. Collectively, our findings indicate that astrocytic Lcn2 contributes to the regulation of the structural plasticity of dendritic spines.

In our study, we have demonstrated that Lcn2 expression increases in astrocytes in vivo during kainate-induced aberrant plasticity. Chia et al. previously demonstrated that Lcn2 expression increases in the hippocampus following intraventricular kainate injection, leading to excitotoxicity [21]. Their study specifically highlighted elevated Lcn2 levels in the CA1 and CA3 regions, where significant neurodegeneration was observed [21]. In turn, our observations show that Lcn2 is increased after kainate in astrocytes, mainly in the DG, a region undergoing gross plastic changes in that model (Figure 1). While we consistently observed a strong increase in Lcn2 expression in the hippocampal DG of the ipsilateral, kainate-treated hemisphere in all analyzed animals, the small sample size precluded statistical inference testing, which would be necessary to quantitatively validate the observed differences in Lcn2 expression.

The localization pattern observed in the kainate experiment suggests a role for astrocytic Lcn2 in plasticity rather than neurodegeneration [42]. To further investigate this plasticity-related function of Lcn2, we employed a chemical long-term potentiation (cLTP) protocol to investigate Lcn2 expression under conditions that induce stable, long-lasting synaptic potentiation in vitro [39]. This protocol combines activation of cAMP-dependent gene expression with a reduction in neuronal inhibition, effectively replicating key aspects of electrically evoked LTP, such as spine enlargement and sustained network activity [43,58]. Our findings have revealed a significant upregulation of Lcn2 expression following cLTP induction, evidenced by an increase in *Lcn2* mRNA levels (Figure 2A) and an elevated number of astrocytes expressing the Lcn2 protein (Figure 2B,C). Interestingly, we observed an increase in the Lcn2 protein level only in a subpopulation of astrocytes. Our data, showing elevated mRNA and protein levels 24 h after stimulation, are in agreement with a recent study by Horino-Shimizu et al., where an increase in Lcn2 expression was also shown 24 h after glutamate stimulation [59]. Additionally, it is possible that astrocytic proteins are more important for memory consolidation rather than expression, which can be supported by the recent study of Sun et al. (2024) [60]. This view was also recently summarized by Murphy-Royal et al. (2023) [61], where the authors argue that astrocytes modulate neuronal networks acting as a contextual gate, which might decode multiple environmental inputs to shape neuronal circuits.

Interestingly, while our results point to a subpopulation of astrocytes as a source of Lcn2 during both kainate-induced aberrant plasticity as well as during cLTP, it is worth noting that the cellular origin of Lcn2 in the brain appears to be context-dependent [21,23,25,28,30,31,32,33]. Various cell types, including neurons, astrocytes, microglia, and endothelial cells, have been identified to express Lcn2 in brain pathologies. After kainate treatment, our immunohistochemistry analysis revealed that most of the Lcn2-positive cells in DG were localized to the hilus, subgranular zone, and molecular layer, with only single cells observed in the granule cell layer, suggesting a non-neuronal origin (Figure 1B). In the following immunofluorescence staining of the DG hilus and granule cell layer, the Lcn2 signal colocalized almost exclusively with the GFAP–astrocytic marker (Figure 1C). In fact, we have not observed the Lcn2 signal, which could not be assigned to GFAP-positive cells. This is in contrast with the massive neuronal Lcn2 expression, which was previously observed after pathological conditions of restrain stress [24,25].

In the basal conditions, there is limited data regarding the expression and cellular origin of Lcn2 in the brain. In our cLTP model, the Lcn2 expression localized to astrocytes (Figure 2B), confirming our in vivo observations. Any residual signal in our immunofluorescence staining comes from autofluorescence induced by 488 laser and was also present in a control staining lacking an anti-Lcn2 primary antibody (Appendix A). Additionally, we were unable to observe if cLTP induces Lcn2 expression in microglia, as our culture model lacks this type of cell [62], but we have not observed the Lcn2-signal that could not be assigned to a GFAP-positive cells. Therefore, future studies should employ more complex approaches, such as triple co-culture systems containing microglia for in vitro experiments and additional microglial and astrocytic markers for in vivo studies, to resolve discrepancies about Lcn2 expression. Our observations indicate that only a subset of astrocytes exhibited increased Lcn2 levels following cLTP, and this subtle, localized increase was challenging to detect via western blotting in vitro (Appendix A) as it is not sensitive enough.

The astrocytic Lcn2 expression in a model of physiological plasticity aligns with the previous findings by Chia et al. (2011) [21], who detected Lcn2 mRNA and protein in several brain regions and demonstrated that, in untreated mice, the Lcn2 protein localizes to astrocytes (olfactory bulb, cerebellum, and brainstem) and to epithelial cells (choroid plexus). Two other studies have also reported low levels of Lcn2 in untreated mice [24,25]. Physiological Lcn2 expression is also supported by a transcriptomic study conducted by Cajigas et al. (2012) [63], who detected *Lcn2* mRNA among transcripts isolated from the neuropil layer of the hippocampal CA1, a brain region that contains dendrites, axons, glia, sparse interneurons, and blood vessels, but not in transcripts from pyramidal neuron somata. Although the specific cellular source of *Lcn2* in their study remains undetermined, these findings are consistent with our observations of astrocytic Lcn2 expression after cLTP. Notably, Lcn2 *mRNA* levels in the Cajigas et al. (2012) [63] study were low (six reads), potentially explaining why we, along with a few other studies, have not detected basal expression of the Lcn2 protein [28,34,64]. However, it is important to note that limited abundance in the physiological brain does not preclude Lcn2 involvement in neuronal plasticity. In fact, low basal expression is characteristic of proteins whose overabundance could disrupt normal brain function, necessitating their tight regulation at multiple levels. For comparison, matrix metalloproteinase-9 (MMP-9), a protein crucial for synaptic plasticity yet implicated in various pathological conditions when overexpressed [65], exhibited similarly low levels (five reads) in the same dataset [63].

One potential mechanism for the regulation of Lcn2 expression during neuronal plasticity could be the cAMP–PKA–CREB pathway that is activated by the cLTP protocol used in our experiments. This pathway has been previously implicated in activity-dependent gene expression in astrocytes [66], and elevated astrocytic cAMP levels in vivo have been shown to trigger synaptic plasticity and influence memory formation [67].

Plasticity-related proteins are not only regulated at the transcriptional and translational levels, but also at the point of their secretion, allowing precisely timed responses to synaptic activity. Using western blotting, we demonstrated rapid Lcn2 release from neuronal–glial co-cultures within 10 min of glutamate stimulation, indicating activity-induced exocytosis (Figure 3). Interestingly, we have not observed a statistically significant increase in the Lcn2–SEP level in the culture medium after 30 min from glutamate stimulation. This is most probably due to endocytosis of the Lcn2 or to the fact that it binds to the cell membrane. The main limitation of the western blot method to detect Lcn2 release to the cell medium is that there is no good reference protein present in the medium, which would not be affected by the stimulation of the culture. To overcome this obstacle, we collected the cell medium before stimulation to show that, in each culture well, the level of Lcn2–SEP was even. Next, the medium was removed and cells were stimulated in a fresh medium to reduce the background signal from Lcn2–SEP accumulated during the culturing period.

Previous studies have shown that, in astrocytes, glutamate evokes a Ca^2+^ concentration rise, leading to the release of gliotransmitters [12,16]. Here, we provided evidence for Ca^2+^-induced Lcn2 exocytosis from pure astrocytic cultures through two complementary approaches. Western blotting revealed increased extracellular Lcn2–SEP levels 20 min after treatment with ionomycin and ATP, both of which are known to elevate intracellular Ca^2+^ [68] (Figure 4). However, this approach has certain limitations. In particular, it is known that astrocytes in pure astrocytic cultures have different morphology and can have different signaling pathways to those present in neuronal–glial co-cultures [66,69,70]. On the other hand, due to low Lcn2 expression and weak sensitivity of the western blot detection method, we used cultures with exogenously expressed Lcn2–SEP. Furthermore, TIRF microscopy confirmed these findings, demonstrating a rapid increase in the Lcn2 release rate within 2 min of ionomycin application (Figure 5). Our results are in agreement with the recent report of Kim et al. (2024) [71], who also demonstrated Lcn2 release from astrocytes. However, the authors applied a different experimental approach [71]. They employed in vivo optogenetic stimulation of channelrhodopsin-2, which induces Ca^2+^ influx in astrocytes, and showed, using microdialysis, a release of Lcn2 after repeated and prolonged stimulation that leads to neuroinflammation. Additionally, the authors used astrocytic cultures and showed a release of Lcn2 24 h after a single 20-minute-long optogenetic stimulation. While their experiments support a general concept of Ca^2+^-induced Lcn2 release, our findings uniquely demonstrate that activity-induced release of Lcn2 can occur within minutes from stimulation. Importantly, other plasticity-related proteins are released at a similar time from stimulation. For instance, the release of neuronal plasticity protein MMP-9 is detectable within seconds after glutamate uncaging when tracked at the level of single dendritic spines [56], and its level in the culture medium increases as early as 5 min after stimulation [72,73]. Additionally, the cleavage of MMP-9 target proteins, such as BDNF and β-dystroglycan, occurs on a similar time scale [72].

Although data on the kinetics of activity-dependent protein release from astrocytes are limited, recent research by Liu et al. (2022) [74] provides valuable insights into the dynamics of BDNF secretion from both neurons and astrocytes and its impact on the establishment of LTP [74]. Their study found that both cell types initiate BDNF release immediately after neuronal stimulation. However, when compared to neuronal BDNF release, astrocytic secretion displayed a smaller peak, longer duration, and slower kinetics. Yet, this specific release dynamics was crucial for the establishment of long-lasting (over 120 min) LTP and remote memory. Our results are consistent with the findings of Liu et al. (2022) [74], as we observed the Lcn2 release immediately after Ca^2+^ influx (within 2 min), and the Lcn2 signal reached high concentrations in the medium within 10–20 min of stimulation. Thus, the dynamics of activity-dependent Lcn2 release align with those observed for other plasticity-related proteins.

Furthermore, we have demonstrated that the 20 min application of exogenous, recombinant Lcn2 significantly increases the length-to-head-width ratio of both mushroom and stubby dendritic spines (Figure 6). This parameter serves as a proxy for spine shape, and its increase suggests spine elongation and/or head shrinkage and thus indicates the transformation of spines toward their less mature, filopodia-like form. These results are in line with the recent studies of Kim et al. (2024) [71], who have shown that the treatment of hippocampal neurons with exogenous Lcn2 reduces levels of surface NMDA receptors, post-synaptic protein PSD levels, and decreases levels of LTP. Our results are also in agreement with Mucha et al. (2011) [25], who reported, in an in vitro model, that prolonged exposure to Lcn2 (3-days) increases the proportion of long spines and decreases the proportion of mushroom spines. However, in vivo studies on the Lcn2 effect on spine morphology show significant variability depending on the brain region examined. For example, Mucha et al. (2011) [25] and Skrzypiec et al. (2013) [24] observed no major changes in baseline spine morphology between wild-type and Lcn2-null genotypes in the CA1, CA3, and basolateral amygdala. By contrast, Ferreira et al. (2013) [22] found that Lcn2-null mice have a decreased proportion of mushroom spines and an increased proportion of long spines in the CA1 of the dorsal hippocampus. In the model of prolonged stress, the impact of Lcn2 on spine morphology also varied by brain region [24,25]. In the CA1 and CA3 fields of the hippocampus, the Lcn2 deletion leads to a stress-induced shift toward mushroom morphology, which is not observed in wild-type animals. Conversely, in the basolateral amygdala, stress-induced changes in spine proportions are generally similar between genotypes [24].

Our results showing the effect of rLcn2 on dendritic spine morphology are, however, in contrast to changes observed after LTP, where synaptic strengthening is associated with dendritic spine enlargement [10]. Interestingly, a similar effect was previously observed for the neuronal plasticity protein MMP-9 [35]. When applied exogenously, MMP-9 promotes an increase in the length/width ratio of dendritic spines [36], and its excessive activity, as observed in transgenic rats, impairs both LTP induction and maintenance [35]. At the same time, other studies have reported a shift toward mushroom-shaped spines during enhanced MMP-9 activity [43,75]. The apparent contradiction was resolved by discovering that, while MMP-9 activity leads to spine elongation, its subsequent inhibition with endogenous inhibitor TIMP-1 is indispensable for spine enlargement and LTP maintenance [35]. Therefore, we conjecture that global application of Lcn2 promotes transition towards long, immature spines and has an adverse effect on LTP, while the transient, local, and controlled action of Lcn2 on synapse is required for spine maturation.

Although our results suggest that Lcn2 acts as a novel activity-dependent molecule released by astrocytes that is involved in neuronal plasticity, our study is not without limitations. First, we show the increase of Lcn2 expression only in a subpopulation of astrocytes within a brain structure. We do not understand what factors regulate this expression and account for the observed diversity. Recent studies have shown that astrocytes, contrary to previous belief, display heterogeneity across different brain regions and even within a region including distinct astrocyte populations in various DG layers [76,77,78,79]. Moreover, while our data support the concept of Ca^2+^-induced Lcn2 release upon activity, the molecular pathways that are activated by glutamate and ATP and result in Lcn2 exocytosis require further investigation. It would be crucial to systematically examine how inhibition of specific components of PLC/IP3 signaling cascade as well as depletion of intracellular Ca^2+^ stores affect the Lcn2 release.

Furthermore, the mechanisms underlying the Lcn2-induced morphological changes are not yet fully understood. One potential mechanism involves Lcn2 interaction with MMP-9. In humans, Lcn2 forms a complex with MMP-9 and regulates its stability and activity [80,81]. This interaction raises an intriguing possibility that astrocytic Lcn2 might influence dendritic spine morphology through its association with neuronal MMP-9, thus providing a novel pathway for astrocytic regulation of LTP. Notably, the Lcn2 release occurs in a time window of MMP-9 action, and Lcn2-induced changes mimic those observed after MMP-9 application [36]. However, it must be stressed that the presence of the Lcn2–MMP-9 complex in the brains of rodents has not yet been confirmed. Lcn2 Cys-87 residue implied in the stabilization of human Lcn2–MMP-9 heterodimer is not present in mice and rats [82,83]. Thus, the involvement of the Lcn2–MMP-9 complex in rodents’ brain plasticity warrants further study. Alternatively, Lcn2 might affect dendritic spines through a mechanism involving iron transfer [25]. Although Lcn2 cannot directly bind iron, it facilitates iron transfer through interaction with siderophore–iron complexes [84,85]. Nevertheless, the specific siderophore–iron complexes bound by Lcn2 in the brain remain unidentified.

## 5. Conclusions

In conclusion, our study supports the role of Lcn2 in the plasticity of dendritic spines and suggests its astrocytic origin in this process. We highlight the importance of precisely regulated Lcn2 release for normal brain function and open new avenues for understanding astrocyte–neuron interactions in learning and memory.

## Figures and Tables

**Figure 1 cells-14-00159-f001:**
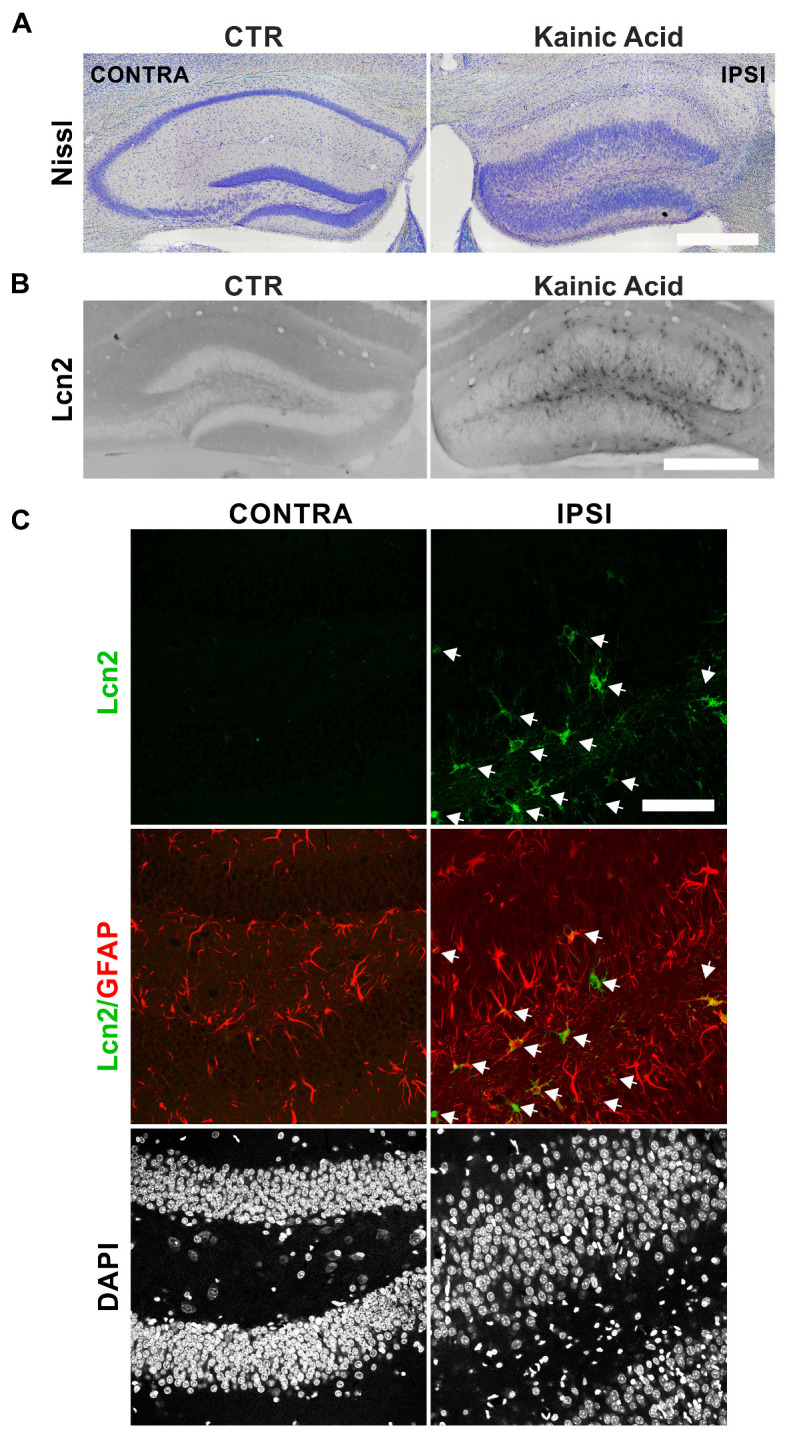
Intrahippocampal injection of KA induces Lcn2 expression in astrocytes in the dentate gyrus. (**A**) A representative example of Nissl staining of the hippocampus of ipsilateral (IPSI) and contralateral sites (CONTRA) of unilaterally KA-injected mice. Scale bar: 500 µm. (**B**) Representative example of immunohistochemical staining of Lcn2 in the hippocampus of control and KA-injected mice. Scale bar: 500 µm. (**C**) Representative example of immunofluorescence images of DG immunostained with antibodies detecting Lcn2 (green) and the astrocytic marker GFAP (red). The nuclei were visualized using DAPI (grey). The Lcn2 protein was observed predominantly in the cytoplasm of GFAP-positive cells. White arrows—Lcn2 and Lcn2/GFAP-positive cells. Scale bar: 5 μm.

**Figure 2 cells-14-00159-f002:**
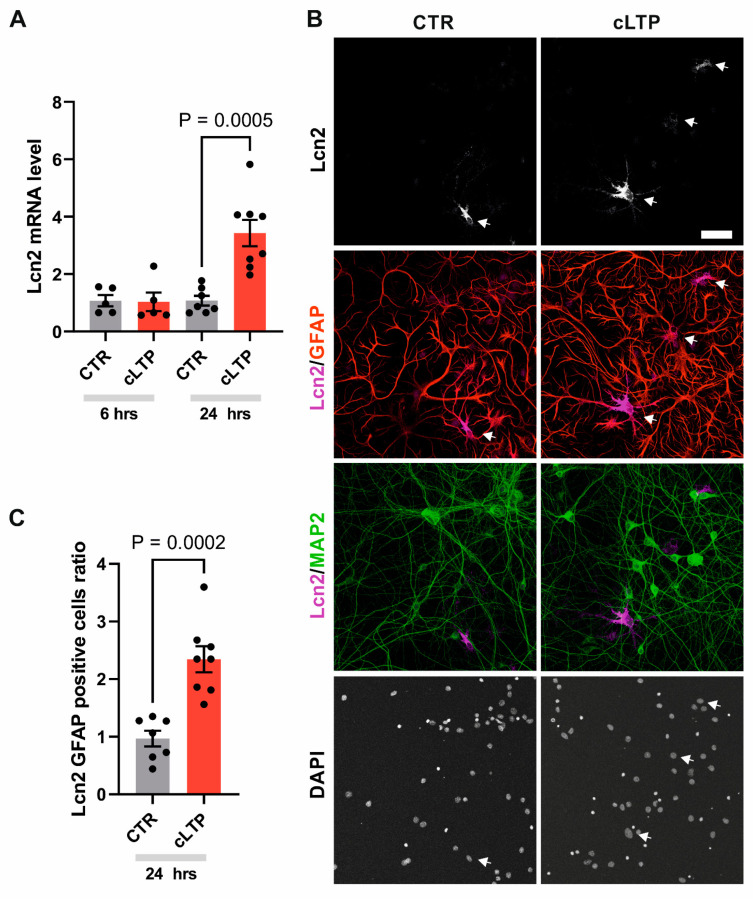
cLTP induces Lcn2 expression in astrocytes of neuronal–glial co-cultures. (**A**) Quantification of real-time RT-PCR analysis of *Lcn2* mRNA levels normalized to the endogenous *Gapdh* in cultures stimulated with cLTP. Bars—means ± SEM; dots—individual measurements of normalized levels of *Lcn2* relative to the level of the non-stimulated control. Student’s *t*-test analysis. (**B**) Representative example of immunofluorescence images of neuronal–glial co-cultures (21 DIV) stimulated with cLTP for 24 h and immunostained with antibodies detecting Lcn2 (white). Middle panels show the co-localization with astrocytic marker GFAP (red) and Lcn2 (magenta), and with the neuronal marker MAP2 (green) and Lcn2 (magenta). The nuclei were visualized using DAPI. The Lcn2 protein was observed almost exclusively in the cytoplasm of GFAP-positive cells. White arrows—Lcn2 and Lcn2/GFAP positive cells. Scale bar: 60 µm. (**C**) Quantification of the number of Lcn2 and GFAP double-positive cells. Bars—means ± SEM; dots—measurements collected from individual coverslips relative to the level of the control. Data were obtained from at least three independent cell cultures. Student’s *t*-test analysis.

**Figure 3 cells-14-00159-f003:**
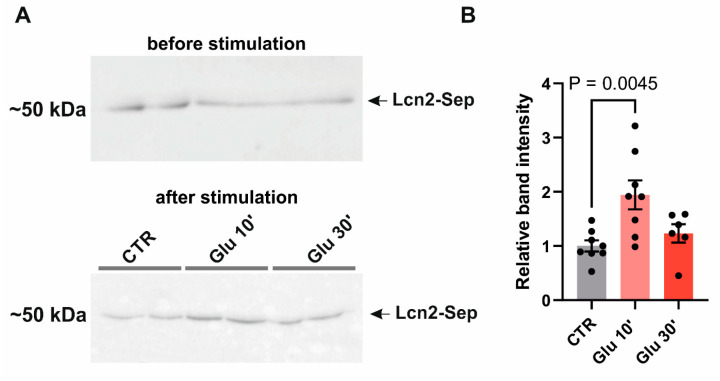
Lcn2 release from neuronal–glial co-cultures is induced by glutamate stimulation. (**A**) Representative immunoblot of Lcn2 (Lcn2–SEP) probed with anti-Lcn2 antibody in the culture media before treatment and in response to 10 and 30 min of 50 µM glutamate (Glu) stimulation. (**B**) Quantification of immunoblot Lcn2–SEP band intensity relative to intensity before stimulation. Data were obtained from at least three independent cell cultures. Bars—means  ±  SEM; dots—individual data points (culture wells). One-way ANOVA, F (2, 19) = 6.580, *p* = 0.0067; Dunnett’s multiple comparisons test CTR vs. 10 min, *p* = 0.0045.

**Figure 4 cells-14-00159-f004:**
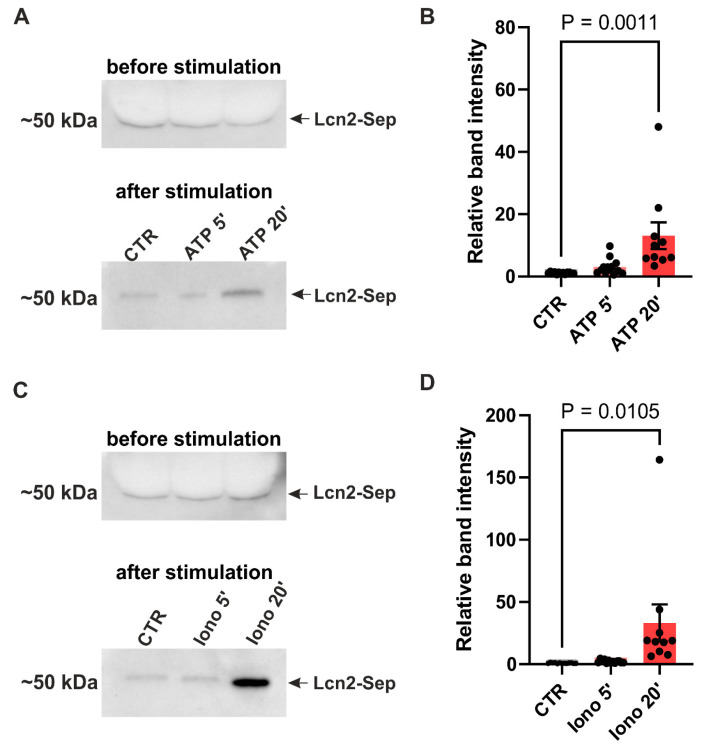
Lcn2 release from astrocytes upon ATP or ionomycin stimulation. (**A**) Representative immunoblot of Lcn2 (Lcn2–SEP) in cell culture media from pure astrocytic culture before and after ATP treatment. (**B**) Quantification of immunoblots of Lcn2–SEP band intensity 5 or 20 min after ATP stimulation relative to the intensity before stimulation. Bars—mean ± SEM; dots—individual data points (media collected from the single culture well). One-way ANOVA, F (2, 33) = 4.230, *p* = 0.0009, Tukey’s multiple comparisons test CTR vs. 20 min ATP, *p* = 0.0011. (**C**) Representative immunoblot of Lcn2 (Lcn2–SEP) in cell culture media from the pure astrocytic culture before and after ionomycin treatment. (**D**) As in (**B**), but for ionomycin stimulation. One-way ANOVA, F (2, 33) = 5.922 *p* = 0.0063, Tukey’s multiple comparisons test CTR vs. 20 min ionomycin *p* = 0.0105. Data was obtained from at least three independent cell cultures.

**Figure 5 cells-14-00159-f005:**
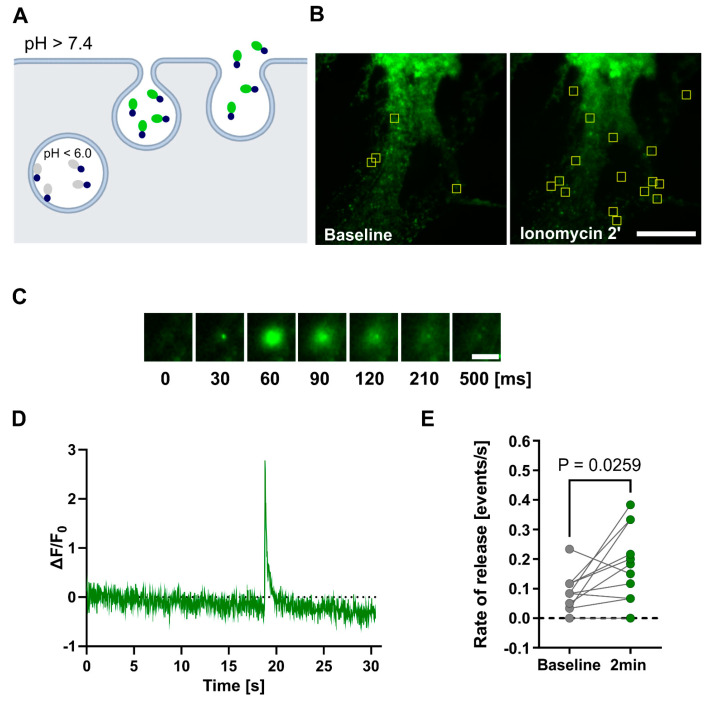
Ca^2+^-dependent Lcn2–SEP release from astrocytes. (**A**) Experiment arrangement: Lcn2 fused to a pH-sensitive fluorescent protein (SEP) is packed into vesicles, where SEP fluorescence is quenched in the acidic environment (pH < 6.0). Upon fusion of the vesicles with the plasma membrane, the pH is neutralized (pH > 7.4), resulting in a sudden increase in fluorescence intensity that can be detected by TIRF microscopy. Black circles—Lcn2; grey ovals—quenched SEP; green ovals—fluorescent SEP. (**B**) Representative TIRF maximal projection of time-stack images shows single-vesicle exocytosis in astrocytes expressing Lcn2–SEP. Localization of Lcn2–SEP exocytotic sites (yellow frames) in pure astrocytic cultures in basal condition or 2 min after ionomycin stimulation. Scale bar: 25 μm. (**C**) Example of the fluorescence increase during a single Lcn2–SEP exocytosis event shown as a sequence of TIRF images acquired over the indicated time. The timescale is relative only for this particular event and shows the temporal resolution of the imaging setup and Lcn2 release. Scale bar: 5 μm. (**D**) Representative example of change in the fluorescence intensity (ΔF/F_0_) for a single exocytosis-positive ROI recorded in astrocytes expressing Lcn2–SEP. The time scale starts with the beginning of the recorded movie. (**E**) Quantification of Lcn2–SEP release rate in response to ionomycin stimulation. Data were obtained from three independent primary astrocytic cultures, dots with lines—individual results for cells in a baseline condition and 2 min after initiating the buffer exchange to the buffer containing ionomycin. Paired student’s *t*-test analysis, *p* = 0.0259, *n* = 11 cells.

**Figure 6 cells-14-00159-f006:**
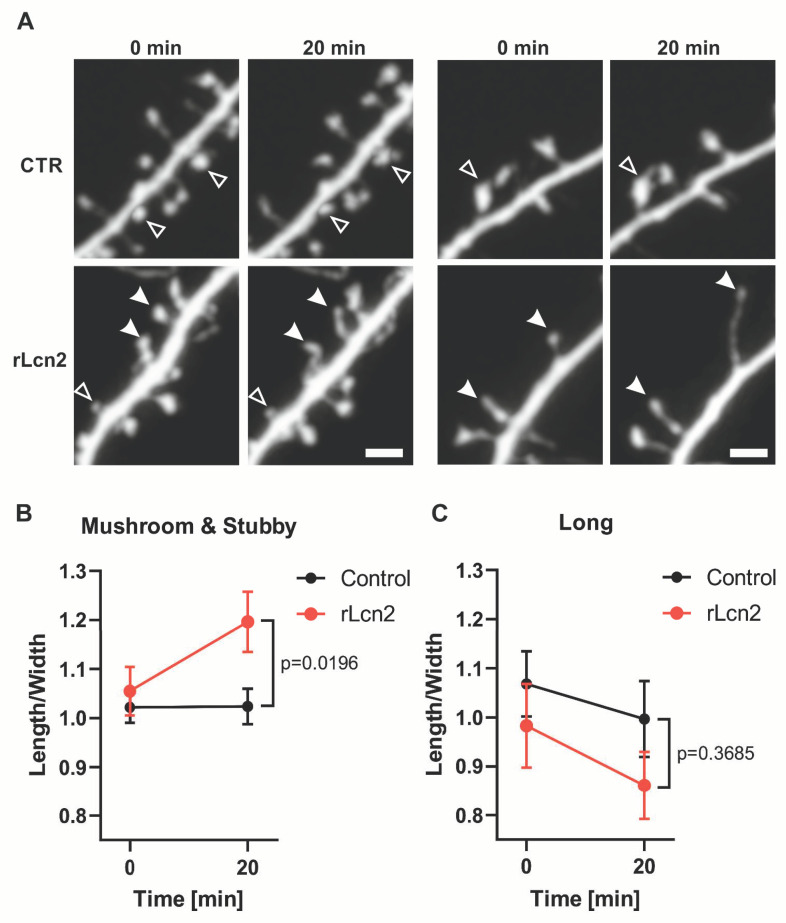
rLcn2 promotes elongation of mushroom and stubby dendritic spines in vitro. (**A**) Two representative examples of GFP-positive dendrites in neuronal–glial co-cultures were acquired during live cell imaging (19–23 DIV) before and after incubation with recombinant Lcn2 (rLcn2) or in control cells. Filled arrowheads show dendritic spines with the increased length-to-width parameter in response to 20 min rLcn2 stimulation, and open arrowheads show an unchanged spine morphology. Scale bar: 2 μm. (**B**) Before and after graph showing the length-to-width parameter of mushroom and stubby dendritic spines incubated with rLcn2 or in the untreated control. The length-to-width of each spine was normalized relative to the average length-to-width of spines in its respective group at time 0. Data were obtained from at least three independent cell cultures. *n* = 14 cells in control and *n* = 13 cells in rLcn2 variant. Repeated measures two-way ANOVA, Time × Lcn2 F (1, 25) = 7.179, *p* = 0.0129; Šídák’s multiple comparisons test control vs. rLcn2 at 20 min, *p* = 0.0196. (**C**) Before and after graph showing the length-to-width parameter of long spines incubated with rLcn2 or in the untreated control. The length-to-width of each spine was normalized relative to the average length-to-width of spines in its respective group at time 0. Data were obtained from at least three independent cell cultures. *n* = 14 cells in control and *n* = 13 cells in rLcn2 variant. Repeated measures two-way ANOVA, time × Lcn2 F (1, 25) = 0.7867, *p* = 0.3835; Šídák’s multiple comparisons test control vs. rLcn2 at 20 min, *p* = 0.3685.

## Data Availability

Raw Data that support the findings of this study are available from the corresponding author upon request.

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
