# Peer review of "Astrocyte-Secreted Lcn2 Modulates Dendritic Spine Morphology"

_cells, 2025, doi:10.3390/cells14030159_

Round 1
Reviewer 1 Report (Previous Reviewer 2)
Comments and Suggestions for Authors
non
Author Response
We are grateful for the evaluation of our paper.
Reviewer 2 Report (New Reviewer)
Comments and Suggestions for Authors
This study explores the role of LCN2 in dendritic spine plasticity, revealing that chemically induced long-term potentiation (cLTP) increases LCN2 expression in astrocytes within mixed hippocampal cultures and that glutamate triggers immediate LCN2 release. While the findings are interesting, several important points require clarification:
1 The authors mention using adult mice but do not specify their age.
2 Why was the experiment conducted 18 days after kainate injection?
3. The authors refer to DAB staining, but the correct term is immunohistochemistry (IHC).
4 Why was the pβactin-LCN2-SEP plasmid created using rat tissue instead of mouse tissue, and why 14 days?
Why were the cultures performed on rat tissue rather than mouse tissue?
i Is the study focused on the hippocampus or cortex? This is unclear.
7 What is the rationale behind performing IHC for LCN2 and Nissl staining?
There is confusion between repeated measures ANOVA and two-way ANOVA. Which test was used, and for which data?
How was the number of LCN2-positive cells quantified?
1How many sections were stained with Nissl and LCN2 IHC?
1Which objective lens was used for cell counting?
1The LCN2 staining image for the control group shows no LCN2-positive astrocytes, despite their normal expression.
1The low number of animals (3-4) limits statistical robustness and result interpretation.
1How was spine morphology analyzed? Length-to-width ratios for mushroom and stubby spines are not comprehensive, especially without data on thin spines.
1What criteria were used to classify spine types?
Author Response
Please see the attachment.

Reviewer 3 Report (New Reviewer)
Comments and Suggestions for Authors
The study presented by Doliwa et al. was investigating the astrocyte derived Lcn-2 on the changes of spine morphology during plasticity with In vitro and in vivo models. The the experimental design is clear and the conclusion is supported by the presented data. I found certain merits of this study to be potentially published here. But some issues need to be addressed during a revision.
1. In both Figure 2 and supplenmatry Figure, the morphological appearances of astrocyte are very weird. Please provide another GFAP staining in normal condition with astrocyte alone.
2. The author showed the data of released protein level of Lcn-2 in medium and the mRNA level in cells. Please provided the western blot results of protein level of Lcn-2 in the cells after c-LTP.
3. In figure 2, the author claimed "Interestingly, we were not able to identify Lcn2 signal in MAP2-positive cells". However, it looked like the Lcn-2 co-localized with both MAP+ neurons and GFAP+ astrocytes if just based on the representative photos.
4. Please explain why there were no internal control in the western blot representative photos.
Round 2
Reviewer 2 Report (New Reviewer)
Comments and Suggestions for Authors
Regarding the comment for staining LCN2, if it needs to improve the staining protocol, it should be done to show positive staining in control group, otherwise the results will not be accurate.
regarding number of animal, n=3,4 is not good enough to run statistical analysis on them and specially parametric test,
the number of sections for quantifications, should be the same for all animals to be able to interpret the results accurately,
since the focus of this study was on the hippocampus even by having similarity between cortex and hippocampus , all analysis should be done on the same region
Author Response
Comments 1: [Regarding the comment for staining LCN2, if it needs to improve the staining protocol, it should be done to show positive staining in control group, otherwise the results will not be accurate.]
Response 1: [Thank you for pointing this out. Despite our efforts, we were not able to clearly detect Lcn2 signal in contralateral hippocampi of kainite-injected mice that would clearly resemble GFAP-positive cells. The level of Lcn2 is simply very low. It does not mean that the fluorescence level is “0”. We can measure the fluorescence level in the outline of GFAP-positive cells in control and kainate-injected hippocampi, however following other comments of the Reviewer, we have decided not to perform statistical analysis of this result. The result clearly shows that Lcn2 levels in ipsilateral hippocampi increases in GFAP-positive astrocytes.]
Comments 2: [Regarding number of animal, n=3,4 is not good enough to run statistical analysis on them and specially parametric test.]
Response 2: [Thank you for pointing this out. Following the Reviewer comments regarding the statistical analysis, we decided to remove statistical analysis from the manuscript and treat the presented data as a qualitative rather than quantitative result. We have changed the Figure 1 and manuscript text accordingly.]
Comments 3: [The number of sections for quantifications, should be the same for all animals to be able to interpret the results accurately.]
Response 3: [Thank you for pointing this out. Following the Reviewer comments regarding the statistical analysis, we decided to remove statistical analysis from the manuscript and treat the presented data as a qualitative rather than quantitative result. We have changed the Figure 1 and the manuscript text accordingly.]
Comments 4: [Since the focus of this study was on the hippocampus even by having similarity between cortex and hippocampus , all analysis should be done on the same region.]
Response 4: [Thank you for pointing this out. Indeed in one of the experiments, we have used mixed cortical cultures of neurons and glia, to show the release of Lcn2. However, for that purpose we have transfected the cells with Lcn2-SEP construct. We do not see how using cortical cultures in this particular experiment can affect the overall observation that Lcn2 is released from astrocytes upon stimulation. We have supported this result by the following experiments using Western-blot analysis of astrocytic culture and TIRF microscopy. Following reviewers concerns, we decided to tone-down the emphasis on hippocampus and changed the manuscript accordingly.]
This manuscript is a resubmission of an earlier submission. The following is a list of the peer review reports and author responses from that submission.
Round 1
Reviewer 1 Report
Comments and Suggestions for Authors
This paper investigates the role of LCN2 in dendritic spine morphology using molecular biological, morphological and TIRF methods, and demonstrates that: 1. Chemically induced long-term potentiation (cLTP) increases Lcn2 expression in astrocytes of mix hippocampal cultures, 2. Instant LCN2 release after glutamate-induced synaptic activation contributes to the morphological changes in dendritic spines, and 3. The calcium influx participates in the LCN2 release in astrocyte.
Overall, this study is interesting as it sheds light into the roles of inflammatory molecular in LTP manipulation as well as dendritic plasticity. However, there are some issues and limitations in the manuscript as well as the figures that need to be addressed and revised. Here are our comments in relation to these issues.
1. The author using immunofluorescence(IF) staining to determine which cell type expresses Lcn2 in response to cLTP, and the results shows that it is GFAP+ astrocyte in stead of MAP2+ Neuron that express LCN2 after cLTP. However, microglia and other cells are also important LCN2 sources under various pathophysiological conditions, and further experiments concerning these cells are needed for LCN2 expression pattern after cLTP.
2. The author uses PCR and IF staining to verified the LCN2 expression after cLTP, and it would be better if extra assays are carried out such as Western Blot (WB) to further quantified the LCN2 changes after cLTP.
3. The authors use WB analysis of culture medium at 10 and 30 min after glutamate treatment, and since the LCN2 is expressed in cells before being secreted out, the LCN2 expression quantifications of culture cell itself could be another strong evidence supporting authors’ opinion. Besides, since Lcn2 protein increased 10 minutes after glutamate while 30 min incubation was not statistically different from not-treated cultures, more time points within 30 mins would be an appropriate way for outlining LCN2 secretion pattern more precisely.
4. The authors use WB assay and the results revealed increased extracellular Lcn2 levels 20 minutes after treatment with Ionomycin and ATP, both of which are known to elevate intracellular Ca2+, which leads to the conclusion that the lcn2 release is inducible by calcium influx, suggesting an activity-dependent release. It is somehow reasonable but the ATP itself participates in LTP as well as dendritic plasticity so that the results of ATP might not sufficient to conclude that ” the lcn2 release is inducible by calcium influx”.
5. Lcn2, in the current study, is showed to be induced by calcium, but what’s the relationship between this calcium-induction manner with lcn2 elevation after cLTP?
6. In discussion part, the authors mention that the cAMP–PKA–CREB pathway could be a potential way of LCN2 regulation during neuronal plasticity. Maybe some molecular-biological experiments could be exerted to further investigate this hypothesis?
7. Some part of the present study could be more plausible if animal studies or gene manipulation techniques are added. For example, the LCN2 knockout in astrocyte could be a better way to further verify the “LTP-LCN2 secretion- dendritic spine morphology change” theory.
Comments on the Quality of English LanguageMinor editing of English language required.
Author Response
1. The author using immunofluorescence (IF) staining to determine which cell type expresses Lcn2 in response to cLTP and the results shows that it is GFAP+ astrocyte in stead of MAP2+ Neuron that express LCN2 after cLTP. However, microglia and other cells are also important LCN2 sources under various pathophysiological conditions, and further experiments concerning these cells are needed for LCN2 expression pattern after cLTP.
Reply: Thank you for your valuable feedback. We agree that microglia and other cell types could be significant sources of Lcn2 under different pathophysiological conditions. We have extended the introduction to provide an overview of Lcn2 cellular sources in the brain during physiological and pathological conditions. However, our study focused on primary hippocampal cell cultures derived from neonatal rats, which were maintained in Neurobasal A medium supplemented with B27 and Glutamax. This standard culture system includes neurons and astrocytes but does not incorporate microglia, as detailed in the reference: Goshi, et al. A primary neural cell culture model to study neuron, astrocyte, and microglia interactions in neuroinflammation. J Neuroinflammation 2020, 17: 155, https://doi.org/10.1186/s12974-020-01819-z.
To investigate the role of microglia in the cLTP model, a more complex experimental setup would be required, such as a triple co-culture model involving neurons, astrocytes, and microglia. This would involve culturing each cell type separately and then combining them in specific ratios to ensure proper integration and functionality. We acknowledge the importance of exploring these additional cell types in the Discussion (page 14 and 15). We appreciate your suggestion and consider it an important direction for future research.
2. The author uses PCR and IF staining to verified the LCN2 expression after cLTP, and it would be better if extra assays are carried out such as Western Blot (WB) to further quantified the LCN2 changes after cLTP.
Reply: Thank you for your suggestion to include Western Blot (WB) analysis to further quantify Lcn2 changes after cLTP. We chose to use immunofluorescence (IF) for detecting changes in Lcn2 protein levels due to its high sensitivity, especially when combined with advanced imaging techniques like confocal microscopy. As demonstrated in Fig. 1, panel B, the increase in Lcn2 expression post-cLTP stimulation is localized to a specific subpopulation of astrocytes. We believe this localized increase would be challenging to detect with Western Blot, which may not capture the spatial distribution of protein expression. Additionally, as mentioned in the manuscript, the endogenous levels of Lcn2 protein are very low. Western Blot analysis for such low-abundance proteins often faces challenges, including higher background noise from nonspecific antibody binding, which can obscure weak signals. For these reasons, we opted for IF as the more suitable method for our study, allowing us to precisely visualize the subtle changes in Lcn2 expression within specific cell populations. We have included a notion in the Discussion of the revised manuscript (page 12) that only a subpopulation of astrocytes seems to increase Lcn2 expression in response to cLTP.
3. The authors use WB analysis of culture medium at 10 and 30 min after glutamate treatment, and since the LCN2 is expressed in cells before being secreted out, the LCN2 expression quantifications of culture cell itself could be another strong evidence supporting authors’ opinion. Besides, since Lcn2 protein increased 10 minutes after glutamate while 30 min incubation was not statistically different from not-treated cultures, more time points within 30 mins would be an appropriate way for outlining LCN2 secretion pattern more precisely.
Reply: Thank you for your valuable suggestions. We appreciate your recommendation to quantify Lcn2 expression within the cell extracts in addition to our analysis of the culture medium. We acknowledge that quantifying intracellular Lcn2 alongside the secreted protein could strengthen our conclusions about the dynamics of Lcn2 secretion in response to glutamate stimulation. In our study, we ensure uniform expression of Lcn2 across wells by utilizing the electroporation-based AMAXA transfection system, which reaches over 80% transfection rate. Both stimulated cells and matching control were seeded from a single post-transfection cell suspension, assuming that this approach would yield uniform transfection efficiency.
Regarding the use of Western Blot (WB), we initially employed this method to preliminarily verify the hypothesis that Lcn2 could be released from cells in response to glutamate stimulation. However, recognizing the limitations of WB where samples are prepared from culture medium and thus lack a legitimate loading control, we used Total Internal Reflection Fluorescence (TIRF) microscopy to more precisely monitor this process. The results from TIRF microscopy, as presented in Figure 4, offer a more straightforward depiction of the dynamics of Lcn2 release. Each of experiments has its own limitations, but we believe that together they provide convincing evidence for the release of Lcn2. Additionally, for Lcn2 signal to be detectable on WB, it needs to accumulate in the medium and that requires time. Therefore, we believe that our TIRF microscopy experiments give quite precise information about the dynamics of Lcn2 release.
4. The authors use WB assay and the results revealed increased extracellular Lcn2 levels 20 minutes after treatment with Ionomycin and ATP, both of which are known to elevate intracellular Ca2+, which leads to the conclusion that the lcn2 release is inducible by calcium influx, suggesting an activity-dependent release. It is somehow reasonable but the ATP itself participates in LTP as well as dendritic plasticity so that the results of ATP might not sufficient to conclude that ”the lcn2 release is inducible by calcium influx”.
Reply: We appreciate the reviewer's point about Ca2+-independent pathways that could be triggered by application of ATP. To clarify, our experiments aiming to determine if the mechanism of Lcn2 exocytosis is Ca2+-induced were carried out in primary astrocytic cultures. This approach allows us to exclude indirect effects due to LTP induction that could be observed upon applying ATP to mixed hippocampal cultures. We selected ATP, in addition to directly increasing intracellular Ca2+ through Ionomycin application, as its action through purinergic receptors is well documented to trigger an increase of Ca2+ in astrocytes through the PLC/IP3 signal cascade. Importantly, ATP has previously been shown to increase Ca2+-level in cultured astrocytes and to induce release of SEP-tagged plasticity protein BDNF in these cultures, see: Stenovec et al. Ketamine inhibits ATP-evoked exocytotic release of brain-derived neurotrophic factor from vesicles in cultured rat astrocytes. Mol Neurobiol 2016, 53: 6882–6896 https://doi.org/10.1007/s12035-015-9562-y. We have highlighted this reasoning in the Results section (page 8).
Importantly, the Western blot results showed that both Ionomycin and ATP stimulation triggered release of Lcn2 from astrocytes. This finding was further corroborated by TIRF data showing Lcn2-release upon Ionomycin treatment. Our conclusion that exocytosis of Lcn2 likely occurs through a Ca2+-induced pathway was based on these combined data. Reassuringly, this conclusion is further supported by recent publication showing release of Lcn2 induced by optogenetic stimulation of channelrhodopsin-2, which causes Ca2+ influx in astrocytes (see: Kim et al. Aberrant Activation of Hippocampal Astrocytes Causes Neuroinflammation and Cognitive Decline in Mice. PLOS Biology 2024, 22: e3002687, doi:10.1371/journal.pbio.3002687). Nevertheless, we agree that we cannot exclude the involvement of other Ca2+-independent pathways triggered by ATP application. We have stressed in the Discussion (page 15) that further studies are needed to better describe the molecular pathway for Lcn2 release. These studies could include the application of intracellular calcium chelators to verify if calcium mobilization is required for Lcn2 exocytosis.
5. Lcn2, in the current study, is showed to be induced by calcium, but what’s the relationship between this calcium-induction manner with lcn2 elevation after cLTP?
Reply: The applied cLTP protocol, which involved the use of forskolin, rolipram, and picrotoxin to stimulate neuronal cultures, leads to a significant rise in network activity, as was indicated by increased spiking and bursting patterns (Szepesi et al., Synaptically Released Matrix Metalloproteinase Activity in Control of Structural Plasticity and the Cell Surface Distribution of GluA1-AMPA Receptors. PLoS One 2014: 9, https://doi.org/10.1371/journal.pone.0098274). Therefore, we expect that it also increases intracellular Ca2+ in astrocytes. Due to application of forskolin and rolipram it also elevates cAMP in these cells. In neurons, simultaneous rise in cAMP and Ca2+-levels leads to activation of CREB transcription factor. We find CREB a possible candidate for a transcription factor responsible for Lcn2 gene regulation in non-inflammatory contexts due to following reasons: 1) neuronal activity has been shown to increase CREB-dependent gene expression in astrocytes (Hasel, et al. Neurons and Neuronal Activity Control Gene Expression in Astrocytes to Regulate Their Development and Metabolism. Nat Commun 2017, 8: 15132, doi: 10.1038/ncomms15132) 2) CREB can regulate Lcn2 expression in the brain in response to stimuli like cocaine (Walker et al. Cocaine Self-administration Alters Transcriptome-wide Responses in the Brain's Reward Circuitry. Biol Psychiatry 2018, 84:867-880. https://doi.org/10.1016/j.biopsych.2018.04.009), 3) Lcn2 mRNA is upregulated by constitutively active CREB in neuronal cultures (Benito et al., cAMP response element-binding protein is a primary hub of activity-driven neuronal gene expression. J Neurosci 2011, 31:18237-50. doi: 10.1523/JNEUROSCI.4554-11.2011). However, it has to be noted that in contrast to neurons, in astrocytes activation of CREB-dependent transcription does not necessarily require elevation of intracellular Ca2+ and cAMP-induced PKA-activation. As an example, it has been reported that ATP and noradrenaline can activate CREB in astrocytes via noncanonical Ca2+-independent pathways that involve atypical protein kinase C (Carriba et al. ATP and noradrenaline activate CREB in astrocytes via noncanonical Ca (2+) and cyclic AMP independent pathways. Glia. 2012, 60: 1330-44. https://doi.org/10.1002/glia.22352). We are therefore cautious in linking increase in intracellular Ca2+ in astrocytes to the increased expression of Lcn2 after cLTP as it is also possible that in astrocytes Lcn2 expression and release are regulated through distinct pathways. We find it an interesting research question and a great direction for future research.
6. In discussion part, the authors mention that the cAMP–PKA–CREB pathway could be a potential way of LCN2 regulation during neuronal plasticity. Maybe some molecular-biological experiments could be exerted to further investigate this hypothesis?
Reply: We agree that identifying molecular cascades affecting Lcn2 expression in astrocytes is an important research question, particularly given that CREB-dependent transcription can also be activated by non-canonical pathways in these cells (Carriba et al. ATP and noradrenaline activate CREB in astrocytes via noncanonical Ca (2+) and cyclic AMP independent pathways. Glia 2012, 60:1330-44. https://doi.org/10.1002/glia.22352); Lim et al. CREB-regulated transcription during glycogen synthesis in astrocytes. Sci Rep 2024, 14: 17942, doi: 10.1038/s41598-024-67976-w). Therefore, further investigation could enhance our understanding of both Lcn2 role in neuronal plasticity and CREB-dependent transcriptional regulation in astrocytes. However, we believe that exploring this hypothesis through molecular-biological experiments extends beyond the scope of the current study, which primarily focuses on the dynamics of LCN2 release in response to stimulation. We appreciate your recommendation and consider it an excellent direction for future research. We plan to explore the pathways regulating Lcn2 expression in astrocytes in subsequent studies.
7. Some part of the present study could be more plausible if animal studies or gene manipulation techniques are added. For example, the LCN2 knockout in astrocytes could be a better way to further verify the “LTP-LCN2 secretion- dendritic spine morphology change” theory.
Reply: Thank you for your thoughtful suggestion. We agree that including animal studies such LCN2 knockout models, could provide deeper insights into the “LTP-LCN2 secretion-dendritic spine morphology change” theory. However, this topic extends beyond the scope of the current manuscript. We consider your suggestion valuable and see it as an important direction for future research.
Reviewer 2 Report
Comments and Suggestions for Authors
This study sought to examine the role of astrocyte-secreted Lipocalin-2 (Lcn2) in neuronal plasticity. The authors of this manuscript reported the following findings: (i) the expression of Lcn2 in cultured hippocampal astrocytes significantly increased 24 hours after treatment by means of a “chemical LTP” protocol; (ii) Lcn2 was released about 20 minutes after application of glutamate, ATP, and ionomycin, respectively; and (iii) brief exposure to recombinant Lcn2 increased the growth of dendritic spines of cultured hippocampal neurons. The authors claimed that Lcn2 is a novel “gliotransmitter” involved in structural plasticity at the tripartite synapse.
Major comments:
1. A more comprehensive introduction would help readers appreciate the current study. For example, it has been reported that inducible Lcn2 is required for stress-induced increase in dendritic spine formation and neuronal activity in rodent brains, and the increased Lcn-2 protein synthesis is predominantly localised to neurons (PLoS One. 2013 Apr 9; 8:e61046). It is also known that Lcn2 is produced and secreted by activated microglia, as well as reactive astrocytes (Experimental & Molecular Medicine 55: 2138–2146; 2023). It is intriguing that the authors of this study were unable to identify Lcn2 signal in MAP2-positive neuronal cells in their cultures. Nevertheless, fluorescence images in the 2nd row of Figure 1B show “small patches” of Lcn2 signal that were not exactly co-localized with neurites of astrocytes. Should the author further analyze the cellular location of these patches of Lcn2 signal that were not associated with GFAP-positive astrocytes?
2. Lcn2 is involved in the control of cell differentiation, energy expenditure, cell death, chemotaxis, cell migration, and many other biological processes. Although six putative receptors for Lcn2 have been proposed, there is a fundamental lack in understanding of how these cell-surface receptors function. (Front Immunol. 2023 Aug 11:14:1229885). Unlike the known gliotransmitters glutamate, D-serine, and ATP, as an acute-phase-response protein Lcn2 is primarily produced and secreted in an inducible fashion. Therefore, one should be cautious of naming Lcn2 as a “novel” gliotransmitter before better understanding of Lcn2 signaling in the central nervous system. In addition, it is inappropriate to conclude that Lcn2 is involved in structure plasticity at the tripartite synapse because only the structure of dendritic spines, but not the presynaptic axonal terminals nor the perisynaptic astrocytic processes, was examined.
3. The authors used a published chemical LTP protocol (cLTP) in this study, trying to indicate a role of astrocyte-produced Lcn2 in LTP-like synaptic plasticity. Yet the cited cLTP protocol (J Neurophysiol 91: 1955–1962, 2004) was used for hippocampal slices (or cultured slices), not for cultured hippocampal cells. In fact, the authors of this study revised the original protocol by adding the LTP chemicals to the cultures after disinhibition of neurons (washing out AMPA/NMDA receptor and voltage-gated Ca2+ channel antagonists). Should the authors carry out specific experiments, for example examining dendritic spine growth, or cite previous studies, to show that this revised “cLTP protocol” did induce LTP responses in cultures of the mix hippocampal cells? In this regards, Lu et al. reported a chemical protocol that induced neuronal LTP in cultured hippocampal cells (Neuron. 2001 Jan; 29(1):243-54), and this cLTP protocol was used in many studies on synaptic plasticity of cultured cortical/hippocampal cells, for example Science 305:1972-1975, 2004.
4. The electrophysiological analyses in the cited cLTP study (J Neurophysiol 91: 1955–1962, 2004) showed that the NMDA receptor-dependent “LTP” of fEPSPs occurred within several minutes after adding the “LTP chemicals” to the hippocampal slices. Also, Lu et al showed that synaptic activities and AMPA receptor insertions to synaptic surface of neurons occurred a few minutes after application of LTP-chemicals (Neuron. 2001 Jan; 29(1):243-54). These studies indicate that the induction of NMDA receptor-dependent chemical LTP takes place rapidly. Indeed, this study showed that dendritic spine growth in neurons increased significantly 20 minutes after adding Lcn2 to the mix hippocampal cultures, indicating that the effect of Lcn2 on neuronal morphology is fast. Of this regard, should the authors explain why they examined the production of Lcn2 in astrocytes 24 hours after applying the cLTP protocol? Did the authors examine the Lcn2 expression level in astrocytes at an earlier time point after applying the cLTP protocol? Or the expression of Lcn2 increased only 24hours after applying the cLTP protocol? These experiments are important for understanding the role of astrocytic Lcn2 in the induction and maintenance of synaptic LTP, or in the regulation of LTP.
5. This reviewer appreciate that this study was aiming to investigate the specific role of astrocyte-secreted Lcn2 in neuronal plasticity. Yet, like neurons and astrocytes microglia also produce and secret Lcn2 and they are critical for synaptic plasticity and learning and memory (Neural Regen Res. 2022 Apr; 17(4): 705–716). Would it be more significant to include microglia in the mix culture thus studying the general role of Lcn2 in synaptic plasticity?
6. In addition to Ca2+ entry, actions of ATP and/or ionomycin in astrocytes are complex. Thus it is not very convincing to use ATP- or ionomycin to induce Lcn2 release thus demonstrating the Ca2+-dependent release of Lcn2 from astrocytes. Hippocampal astrocytes express Ca2+-permeable AMPA and NMDA receptors (Biomolecules. 2021 Oct; 11: 1467). Testing the effects of AMPA/NMDA on Lcn2 secretion from astrocytes in Ca2+-free medium (for about 10 min) might be a better experiment.
Minor comments:
1. The “chemical LTP protocol” was used before the cited study (J Neurophysiol 91: 1955–1962, 2004). The authors should cite the original study.
2. The immunoblot analysis of Lcn2 in Figure 2A and in Figure 3A-B should have a loading control, and label the molecular weight of the “Lcn2” band.
3. It is intriguing that the Lcn2 secretion increased in 10 min after 50μm glutamate stimulation but decreased 30 min after glutamate stimulation. The author should discuss this phenomenon.
4. The relationship of times course in Figure 4C and 4D should be clearly described in the figure legend. This reviewer was confused by the brightest fluorescence in a single exocytosis at about 50 ms (4C) vs the peak of fluorescence intensity in a single exocytosis at about 20 min (4D).
5. Should “Life imaging” be “Live cell imaging”?
Comments on the Quality of English LanguageThe quality of writing is good.
Author Response
1. A more comprehensive introduction would help readers appreciate the current study. For example, it has been reported that inducible Lcn2 is required for stress-induced increase in dendritic spine formation and neuronal activity in rodent brains, and the increased Lcn-2 protein synthesis is predominantly localised to neurons (PLoS One. 2013 Apr 9; 8:e61046). It is also known that Lcn2 is produced and secreted by activated microglia, as well as reactive astrocytes (Experimental & Molecular Medicine 55: 2138–2146; 2023). It is intriguing that the authors of this study were unable to identify Lcn2 signal in MAP2-positive neuronal cells in their cultures. Nevertheless, fluorescence images in the 2nd row of Figure 1B show “small patches” of Lcn2 signal that were not exactly co-localized with neurites of astrocytes. Should the author further analyze the cellular location of these patches of Lcn2 signal that were not associated with GFAP-positive astrocytes?
Reply: We thank the reviewer for that comment. We are aware of differences in Lcn2 localization found in different studies but our intention was not to stretch the introduction as this is an experimental paper not a review. However, following your comment, we have now provided a more comprehensive overview of Lcn2 cellular sources in the brain during physiological and pathological conditions, as well as about inducible manner of Lcn2 expression in brain pathologies (page 2). Indeed, it is surprising that we didn’t find the localization in neurons; however, there are limited publications showing basal Lcn2 expression in the brain, and as stated in the Discussion, the available data (Chia et al. Expression and Localization of the Iron-Siderophore Binding Protein Lipocalin 2 in the Normal Rat Brain and after Kainate-Induced Excitotoxicity. Neurochem Int 2011, 59: 591–599, doi:10.1016/j.neuint.2011.04.007; Cajigas, et al. The Local Transcriptome in the Synaptic Neuropil Revealed by Deep Sequencing and High-Resolution Imaging. Neuron 2012, 74: 453–466, doi: 10.1016/j.neuron.2012.02.036) are in line with our observations of Lcn2 expression in astrocytes. The neuronal localization of Lcn2 was shown after prolonged psychological stress (Skrzypiec et al. Stress-Induced Lipocalin-2 Controls Dendritic Spine Formation and Neuronal Activity in the Amygdala. PLoS One 2013, 8, doi: 10.1371/journal.pone.0061046; Mucha et al. Lipocalin-2 Controls Neuronal Excitability and Anxiety by Regulating Dendritic Spine Formation and Maturation. PNAS 2011, 108: 18436–18441, doi: 10.1073/pnas.1107936108) but these publications do not present data on the cellular localization of Lcn2 under physiological conditions, which we could discuss in detail.
Indeed, Lcn2 has been reported to be expressed in microglia, yet, mix hippocampal cultures kept in Neurobasal/B27 medium are usually devoid of microglia and Iba1 staining (marker of microglia) was always negative in our cultures. A recent paper of Goshi et al. A primary neural cell culture model to study neuron, astrocyte, and microglia interactions in neuroinflammation. J Neuroinflammation 2020, 17: 155, https://doi.org/10.1186/s12974-020-01819-z shows that mix cultures need to be maintained in a medium supplemented with IL-34, TGF-b and ovine wool cholesterol, which was not our case. Therefore, we excluded the possibility that Lcn2 signal comes from microglia. Nevertheless, we have highlighted in the discussion that source of Lcn2 in the brain appears to be context-dependent (page 12) and that further in vitro studies using more complex triple co-culture systems that contain microglia, as well as in vivo studies using techniques that increase the sensitivity of detection, such as tyramide signal amplification, could be helpful to delineate existing discrepancies about basal Lcn2 expression (page 14 and 15).
The “small patches” in Lcn2 signal are nonspecific signal caused by auto-fluorescence of hippocampal culture which is often observed in 488 channel. However, the “patches” are much weaker than specific signal which clearly colocalizes with GFAP signal. We have now added a Supplementary figure showing that this background is also present in the negative control lacking primary anti-Lcn2 antibody and added an appropriate comment in Discussion (page 13). GFAP signal comes from intermediate filament and does not fill the entire cytoplasm of an astrocyte. It is rather concentrated in the cell soma and proximal branches. It has been previously demonstrated that GFAP occupies only ∼15% of the total astrocyte volume (Bushonget al. Protoplasmic astrocytes in CA1 stratum radiatum occupy separate anatomical domains. J Neurosci 2002, 22: 183-92, doi: 10.1523/JNEUROSCI.22-01-00183.2002) Therefore, it is not surprising that Lcn2 signal is not the same as GFAP.
2. Lcn2 is involved in the control of cell differentiation, energy expenditure, cell death, chemotaxis, cell migration, and many other biological processes. Although six putative receptors for Lcn2 have been proposed, there is a fundamental lack in understanding of how these cell-surface receptors function. (Front Immunol. 2023 Aug 11:14:1229885). Unlike the known gliotransmitters glutamate, D-serine, and ATP, as an acute-phase-response protein Lcn2 is primarily produced and secreted in an inducible fashion. Therefore, one should be cautious of naming Lcn2 as a “novel” gliotransmitter before better understanding of Lcn2 signaling in the central nervous system. In addition, it is inappropriate to conclude that Lcn2 is involved in structure plasticity at the tripartite synapse because only the structure of dendritic spines, but not the presynaptic axonal terminals nor the perisynaptic astrocytic processes, was examined.
Reply: We thank the reviewer for that comment and agree that the term: “novel gliotransmitter” is an overstatement. We corrected the manuscript accordingly. We are sorry that the sentence: “…involved in structural plasticity at the tripartite synapse” was confusing, we meant that if one of the components of the tripartite synapse changes (in this case postsynaptic part) therefore the tripartite synapse is changed. We have changed the sentence and made it more precise.
3. The authors used a published chemical LTP protocol (cLTP) in this study, trying to indicate a role of astrocyte-produced Lcn2 in LTP-like synaptic plasticity. Yet the cited cLTP protocol (J Neurophysiol 91: 1955–1962, 2004) was used for hippocampal slices (or cultured slices), not for cultured hippocampal cells. In fact, the authors of this study revised the original protocol by adding the LTP chemicals to the cultures after disinhibition of neurons (washing out AMPA/NMDA receptor and voltage-gated Ca2+ channel antagonists). Should the authors carry out specific experiments, for example examining dendritic spine growth, or cite previous studies, to show that this revised “cLTP protocol” did induce LTP responses in cultures of the mix hippocampal cells? In this regards, Lu et al. reported a chemical protocol that induced neuronal LTP in cultured hippocampal cells (Neuron. 2001 Jan; 29(1):243-54), and this cLTP protocol was used in many studies on synaptic plasticity of cultured cortical/hippocampal cells, for example Science 305:1972-1975, 2004.
Reply: We thank the reviewer for the comment. Indeed, the original protocol was used in hippocampal slices, but we and others have successfully used it to induce LTP in dissociated hippocampal cultures in many previously published studies. We have corrected the Results section (page 6) and provided appropriate references.
4. The electrophysiological analyses in the cited cLTP study (J Neurophysiol 91: 1955–1962, 2004) showed that the NMDA receptor-dependent “LTP” of fEPSPs occurred within several minutes after adding the “LTP chemicals” to the hippocampal slices. Also, Lu et al showed that synaptic activities and AMPA receptor insertions to synaptic surface of neurons occurred a few minutes after application of LTP-chemicals (Neuron. 2001 Jan; 29(1):243-54). These studies indicate that the induction of NMDA receptor-dependent chemical LTP takes place rapidly. Indeed, this study showed that dendritic spine growth in neurons increased significantly 20 minutes after adding Lcn2 to the mix hippocampal cultures, indicating that the effect of Lcn2 on neuronal morphology is fast. Of this regard, should the authors explain why they examined the production of Lcn2 in astrocytes 24 hours after applying the cLTP protocol? Did the authors examine the Lcn2 expression level in astrocytes at an earlier time point after applying the cLTP protocol? Or the expression of Lcn2 increased only 24hours after applying the cLTP protocol? These experiments are important for understanding the role of astrocytic Lcn2 in the induction and maintenance of synaptic LTP, or in the regulation of LTP.
Reply: We thank the reviewer for this insightful question. It is true that most studies of LTP-induced expression focus on much earlier time points (mostly two or six hours) after the stimulation, however, we do not think it must be the case for astrocytes. Indeed, we have made a pilot experiment on a small sample and checked for Lcn2 expression 3h, 6h after cLTP but we didn’t see any increase in Lcn2 expression. That’s why we have decided to check expression after 24h on a larger sample. It is possible, that astrocytic proteins are more important for memory consolidation rather than memory expression, which can be supported by the recent study of Sun et al. Spatial transcriptomics reveal neuron–astrocyte synergy in long-term memory. Nature 2024, 627: 374–381, doi: https://doi.org/10.1038/s41586-023-07011-6. This view was summarized recently in the following paper: Murphy-Royal et al. A conceptual framework for astrocyte function. Nat Neurosci 2023, 26: 1848–1856, https://doi.org/10.1038/s41593-023-01448-8. Additionally, our results are in agreement with a recent paper by Horino-Shimizu et al. Lipocalin-2 production by astrocytes in response to high concentrations of glutamate. Brain Res. 2023, 1815: 148463, https://doi.org/10.1016/j.brainres.2023.148463. The authors showed increase in Lcn2 expression (on both mRNA and protein level) 24h after glutamate stimulation. We have added appropriate paragraph in the Discussion.
5. This reviewer appreciate that this study was aiming to investigate the specific role of astrocyte-secreted Lcn2 in neuronal plasticity. Yet, like neurons and astrocytes microglia also produce and secret Lcn2 and they are critical for synaptic plasticity and learning and memory (Neural Regen Res. 2022 Apr; 17(4): 705–716). Would it be more significant to include microglia in the mix culture thus studying the general role of Lcn2 in synaptic plasticity?
Reply: We thank the reviewer for this suggestion. Yes, we agree that studying microglia together with neurons and astrocytes is crucial for proper understanding of Lcn2 action, however, we believe that it would require conducting experiments far exceeding the scope of this study. It is however a great idea for future projects. As we have already mentioned answering the question 1, we do not exclude the possibility that microglia release Lcn2, but as we stated, our experimental model is devoid of this type of cells. We added the Discussion section (page 14 and 15) pointing limitations of our study and encouraging further in vitro studies using more complex triple co-culture systems that contain microglia.
6. In addition to Ca2+ entry, actions of ATP and/or ionomycin in astrocytes are complex. Thus it is not very convincing to use ATP- or ionomycin to induce Lcn2 release thus demonstrating the Ca2+-dependent release of Lcn2 from astrocytes. Hippocampal astrocytes express Ca2+-permeable AMPA and NMDA receptors (Biomolecules. 2021 Oct; 11: 1467). Testing the effects of AMPA/NMDA on Lcn2 secretion from astrocytes in Ca2+-free medium (for about 10 min) might be a better experiment.
Reply: We thank the reviewer for that comment. Indeed, ATP action on astrocytes might be considered complex, as there are two main types of ATP receptors – ionotropic P2X and metabotropic P2Y. However, many P2X receptors and P2Y receptors lead to increase in cytosolic Ca2+ but from different sources – extracellular vs intracellular stores. Furthermore, we find ATP physiologically relevant stimuli to study Ca2+-dependent exocytosis as at least in astrocytic cultures it is involved in propagation of Ca2+- waves between astrocytes (Bowser DN and Khakh BS. Vesicular ATP is the predominant cause of intercellular calcium waves in astrocytes. J Gen Physiol 2007, 129: 485-91. doi: 10.1085/jgp.200709780). In case of ionomycin, we are not aware of controversies which would show different mode of action then increase in cytosolic Ca2+ concentration. It’s true that some reports point to complexity of Ca2+ elevations in response to ionomycin treatment. However, in the presence of 2 mM extracellular Ca2+ concentration which is high in relation to basal cytosolic Ca2+ concentration (as was the case of our experiments), ionomycin treatment leads to rapid increase in cytosolic Ca2+ levels within seconds from the onset of exposure. See for example: Müller et al. Complex Actions of Ionomycin in Cultured Cerebellar Astrocytes Affecting Both Calcium-Induced Calcium Release and Store-Operated Calcium Entry. Neurochem Res 2023, 38: 1260–1265, https://doi.org/10.1007/s11064-013-1021-4. As we already stated in response to Reviewer 1, we argue that combined data from ATP and Ionomycin treatment suggest that exocytosis of Lcn2 is Ca2+-inducible. This conclusion is in line with recent data from Kim et al. 2024, (Aberrant Activation of Hippocampal Astrocytes Causes Neuroinflammation and Cognitive Decline in Mice. PLOS Biology 2024, 22: e3002687, doi:10.1371/journal.pbio.3002687) showing release of Lcn2 induced by optogenetic stimulation of channelrhodopsin-2, which causes Ca2+ influx in astrocytes. Nevertheless, we agree that based on our data, we cannot exclude the involvement of other Ca2+-independent pathways triggered by ATP application. We have stressed in the Discussion (page 15) that further studies are needed to better describe the molecular pathway for Lcn2 release. These studies could include the application of intracellular calcium chelator to verify if calcium mobilization is required for Lcn2 exocytosis.
Minor comments:
1. The “chemical LTP protocol” was used before the cited study (J Neurophysiol 91: 1955–1962, 2004). The authors should cite the original study.
Reply: We thank the reviewer for that comment, however we are not aware of a study which had used cLTP protocol consisting of Forskolin, Rolipram and Picrotoxin, before Otmakhov N et al. Forskolin-Induced LTP in the CA1 Hippocampal Region Is NMDA Receptor Dependent. Journal of Neurophysiology 2004, 91, 1955–1962, doi: 10.1152/jn.00941.2003. Previously published LTP-inducing protocols used rather Sp-cAMPS (an analog of cAMP) for example Frey et al. Effects of cAMP simulate a late stage of LTP in hippocampal CA1 neurons. Science 1993 260: 1661–1664, doi: 10.1126/science.8389057, Bolshakov et al. Recruitment of new sites of synaptic transmission during the cAMP-dependent late phase of LTP at CA3-CA1 synapses in the hippocampus. Neuron 1997, 19: 635–651, doi: 10.1016/s0896-6273(00)80377-3 and others. It is also true that there are different cLTP protocols (already mentioned by the reviewer) but they base on 0 mM Mg2+ and glycine to stimulate NMDAR, but we have not used those protocols. There are also protocols which combine for example Forskolin and 0mM Mg2+ like in a recent paper of Bimbi and Tongiorgi, Chemical LTP induces confinement of BDNF mRNA under dendritic spines and BDNF protein accumulation inside the spines. Front Mol Neurosci 2024, 17: 1348445, https://doi.org/10.3389/fnmol.2024.1348445 .
2. The immunoblot analysis of Lcn2 in Figure 2A and in Figure 3A-B should have a loading control, and label the molecular weight of the “Lcn2” band.
Reply: We thank the reviewer for that comment. The samples analysed by Western-Blot, were prepared from culture medium. We are not aware of a loading control in such a case – i.e. a protein constitutively released to the cellular medium. We have taken extreme care to load the same amount of a sample to each well and we repeated the experiment three times. We labeled the molecular weight of the “Lcn2” band in the Fig. 2A and Fig.3 A, B.
3. It is intriguing that the Lcn2 secretion increased in 10 min after 50μm glutamate stimulation but decreased 30 min after glutamate stimulation. The author should discuss this phenomenon.
Reply: We thank the reviewer for that comment. We have added appropriate comments in the revised manuscript. The samples were prepared from culture medium and Lcn2 in later times i.e. 30 min after stimulation, might be bound to the cellular membrane or simply removed from the medium through endocytosis.
4. The relationship of times course in Figure 4C and 4D should be clearly described in the figure legend. This reviewer was confused by the brightest fluorescence in a single exocytosis at about 50 ms (4C) vs the peak of fluorescence intensity in a single exocytosis at about 20 min (4D).
Reply: We thank the reviewer for that comment and we are sorry for not making the figure caption clearer. We have revised the manuscript to improve clarity.
5. Should “Life imaging” be “Live cell imaging”?
Reply: We thank the reviewer for that comment. Indeed ”Life cell imaging” is more clear and we have changed the manuscript accordingly.
Round 2
Reviewer 1 Report
Comments and Suggestions for Authors
Thanks to the authors for the detailed and meticulous reply. The authors are very familiar with the relevant field of their study and the references cited in this manuscript are appropriate and relevant to this research. However, I believed there still exists several problems that need to be revised.
1. It would be better if the revision made concerning LCN2 cellular sources in the brain during physiological and pathological conditions and why the present study only focused on astrocyte and neuron were placed, which was pleasing to read and was very well written, in the discussion section.
2. The authors said that “We believe this localized increase would be challenging to detect with Western Blot, which may not capture the spatial distribution of protein expression”, then why using PCR for LCN2 level detection which faced the same problem? It is correct that the WB or ELISA mentioned in the first comment report is not essential in the study, but only one kind of experiment such as IF or TIRF is not sufficient for the conclusion the authors made. It would be pleasing if the authors take all the response seriously instead of making plausible excuses.
3. According to the results presented, the current study showed that:1.the LCN2 elevates in the astrocyte after cLTP, glutamate or calcium usage, and 2.the LCN2 participate substantially in the dendritic spine morphology. The potential molecular mechanism beneath is not tested in the present study but only in discussion part. Therefore, the current study unveiled interesting phenomena without further mechanism investigation, which lower the readers' interest
Reviewer 2 Report
Comments and Suggestions for Authors
1) By any means, immunoblot analyses should have a loading control although it is complex for soluble proteins. The authors may consider soaking the gel with Coomassie blue to show the "total" proteins in the gel.
2) Live cell imaging